# Assessing the response of soil carbon in Australia to changing inputs and climate using a consistent modelling framework

Juhwan Lee[1], Raphael A. Viscarra Rossel[1], Mingxi Zhang[1], Zhongkui Luo[2], and Ying Ping Wang[3]

[1]Soil & Landscape Science, School of Molecular and Life Sciences, Curtin University, GPO Box U1987, Perth WA 6845, Australia.
[2]College of Environmental and Resource Sciences, Zhejiang University, Hangzhou, Zhejiang, China.
[3]CSIRO Oceans and Atmosphere, Private Bag 1, Aspendale, VIC 3195, Australia.
**Correspondence:** Raphael A. Viscarra Rossel (r.viscarra-rossel@curtin.edu.au)

**Abstract.** Land use and management practices affect the the response of soil organic carbon (C) to global change. Process-based models of soil C are useful tools to simulate C dynamics, but it is important to bridge any disconnect that exists between the data used to inform the models and the processes that they depict. To minimise that disconnect, we developed a consistent modelling framework that integrates new spatially-explicit soil measurements and data with the Rothamsted carbon model (ROTH C) and simulated the response of soil organic C to future climate change across Australia. We compiled publicly available continental-scale datasets and pre-processed, standardised and configured them to the required spatial and temporal resolutions. We then calibrated ROTH C and run simulations to estimate the baseline soil organic C stocks and composition in the 0–0.3 m layer at 4,043 sites in cropping, modified grazing, native grazing, and natural environments across Australia. We used data on the C fractions, the particulate, mineral-associated and resistant organic C (POC, MAOC and ROC, respectively) to represent the three main C pools in the ROTH C model's structure. The model explained 97–98% of the variation in measured total organic C in soils under cropping and grazing, and 65% in soils under natural environments. We optimised the model at each site and experimented with different amounts of C inputs to simulate the potential for C accumulation under constant climate in a 100-year simulation. With an annual increase of 1 Mg C ha$^{-1}$ in C inputs, the model simulated a potential soil C increase of 13.58 (interquartile range 12.19–15.80), 14.21 (12.38–16.03), and 15.57 (12.07–17.82) Mg C ha$^{-1}$ under cropping, modified grazing and native grazing, and 3.52 (3.15–4.09) Mg C ha$^{-1}$ under natural environments. With projected future changes in climate (+1.5°, 2° and 5.0°C) over a 100-years, the simulations showed that soils under natural environments lost the most C, between 3.1 and 4.5 Mg C ha$^{-1}$, while soils under native grazing lost the least, between 0.4 and 0.7 Mg C ha$^{-1}$. Soil under cropping lost between 1 and 2.7 Mg C ha$^{-1}$, while those under modified grazing showed a slight increase with temperature increases of 1.5°C, but with further increases of 2° and 5°C the median loss of TOC was 0.28 and 3.4 Mg C ha$^{-1}$, respectively. For the different land uses, the changes in the C fractions varied with changes in climate. An empirical assessment of the controls on the C change showed that climate, pH, total N, the C:N ratio, and cropping were the most important controls on POC change. Clay content and climate were dominant controls on MAOC change. Consistent and explicit soil organic C simulations improve confidence in the model's estimations, facilitating the development of sustainable soil management under global change.

## 1 Introduction

Soil carbon (C) represents the most abundant terrestrial C pool (Batjes, 1996). It can be a significant source, or sink of atmospheric $CO_2$ (Scharlemann et al., 2014). Sequestration of soil organic C, via the adoption of innovative land management strategies, offers opportunities for improving soil and ecosystem health, sustaining food production and mitigating climate change (Lal, 2016; Paustian et al., 2019; Smith et al., 2020). However, these opportunities depend on regional interactions between soil, climate, land use and management (Viscarra Rossel et al., 2019). A better understanding of the effect of these interactions on soil C is needed to assess the potential for those opportunities.

Biogeochemical models represent our mechanistic understanding of processes such as organic C cycling in soil and can serve different purposes. They can be used to simulate soil C cycling under various combinations of soil, climate, land use and management (Conant et al., 2011), to evaluate the potential for C sequestration or loss, and to assess the impacts of environmental and human-induced change in the soil C cycle. In conjunction with long-term measurements, models can estimate the effects of management practices and climate change on soil C, as well as subsequent feedbacks. Therefore, the simulation of soil organic C with biogeochemical models has received much attention in the literature (Campbell and Paustian, 2015; Falloon and Smith, 2000).

The Rothamsted carbon model (ROTH C) (Jenkinson, 1990; Coleman and Jenkinson, 1996) and the CENTURY model (Parton et al., 1987) are widely used to simulate soil organic C dynamics in cropping, grassland and forest systems. Although developed under northern hemisphere conditions, since their inception in the 1980s, these models have been used for many different applications worldwide (Campbell and Paustian, 2015; Wang et al., 2016). They are the soil biogeochemical component in Earth systems models (Todd-Brown et al., 2013). They do not explicitly represent current theories around the mechanisms of microbial decomposition and physicochemical protection (Lehmann and Kleber, 2015), but they are still being used because they capture the general principle of soil organic C dynamics. In essence, the flow of C in the models occur through a cascading of C via several conceptual pools turning over at different rates, according to first-order kinetics and modified by climate and soil texture. Other reasons for their continued use might be that there is ample documentation on them; they are relatively robust and well tested.

The ROTH C model has been adjusted and tested for use under Australian conditions (Janik et al., 2002; Skjemstad et al., 2004). Skjemstad et al. (2004) showed that the size of the main conceptual C pools in ROTH C, the resistant plant material, humic and inert organic matter pools, can be initialised with measurements of the particulate, mineral-associated and resistant organic C fractions (POC, MAOC and ROC, respectively). ROTH C can be initialised with measured C fractions. Skjemstad et al. (2004) calibrated the decomposition rate constants under Australian conditions, and Janik et al. (2002) assessed a sensitivity of the C pools to model parameters to highlight the potential complexity in the implementation of ROTH C. Since then, researchers in Australia have used ROTH C in different studies (e.g Paul and Polglase, 2004; Lee and Viscarra Rossel, 2020). ROTH C is a sub-model of the Fully Integrated Carbon Accounting Model (FullCAM) (Richards and Evans, 2004), used in Australia's National Greenhouse Gas Inventory System. Together, they are the core of the Australian model-based Emission Reduction Fund (ERF) methodology, which allows farmers and landholders to generate extra income by storing C in their soils

and thereby reducing emissions (England and Viscarra Rossel, 2018; Paustian et al., 2019). However, the soil and environmental conditions required to maintain current soil organic C stocks and composition are poorly understood. This hampers the reliable estimation of C stock and sequestration potentials at a large scale.

Despite the development of new models with updated representations of current understanding (Abramoff et al., 2018; Robertson et al., 2019; Wieder et al., 2014), there remains a disconnect between measurements and datasets used to inform the models and the theories represented in them, particularly for simulations over large extents (Blankinship et al., 2018; Harden et al., 2018). In practice, a lack of data restricts model parameterisation and optimisation, and missing temporal datasets limit our ability to simulate and verify long-term changes in soil C stocks and composition (Smith et al., 1997). Hence, there is also little agreement on how input datasets should be synthesised, processed and used (Manzoni and Porporato, 2009), leading to inconsistent model calibrations (Conant et al., 2011; Seidel et al., 2018) and inaccurate model estimations (Shi et al., 2018). In this context, the development of robust frameworks for soil organic C modelling and simulation, to synthesise and integrate measurements and datasets with models are critical (Harden et al., 2018; Ogle et al., 2010; Paustian et al., 1997; Smith et al., 2020). Their development should also allow for their efficient updating, with new measurements, data and models, as they become available (Viscarra Rossel and Brus, 2018; England and Viscarra Rossel, 2018; Smith et al., 2020), and enable a more systematic approach for calibration and validation, making simulations more reliable and reproducible.

Here, we report on simulations of the organic C stocks in Australian soils with ROTH C using a standardised approach that synthesises and processes measurements and data for prediction at the required scale. Our motivation for developing this research is to help answer questions around soil C dynamics that are pertinent to Australian soils in different ecosystems and under different land uses and management. Our aims are to: (i) derive baseline estimates of soil organic C stock and composition by site-specifically initialising the model with measurements of POC, MAOC and ROC and an optimised ratio of decomposable plant material (DPM) to resistant plant material (RPM), which represents the decomposability of incoming biomass, (ii) simulate over a 100-year period, with constant and changing climate and a plausible range of C inputs, the potential to increase organic C stocks as well as the potential vulnerability to C loss across Australia, and (iii) to identify the soil and environmental controls of the change in soil C stocks.

## 2   Materials and methods

### 2.1   The Rothamsted carbon model (ROTH C)

ROTH C is a soil process model for the turnover of organic C in non-flooded soils (Jenkinson, 1990; Coleman and Jenkinson, 1996). The model partitions total organic C (TOC) into pools that represent decomposable plant material (DPM), resistant plant material (RPM), microbial biomass (BIO), humified organic matter (HUM), and inert organic matter (IOM) (Coleman and Jenkinson, 1996). The model simulates on a monthly time step changes in its active pools, in response to climate, soil type, land use and management. Annual C inputs from crops and manure represent different land use and management regimes. We used the ROTH C model version 26.3, which is the version that was re-calibrated for a range of Australian soils (Skjemstad et al., 2004). The decomposition rate constants for the DPM, BIO, RPM, and HUM pools are 10, 0.66, 0.15, and 0.02 year$^{-1}$,

respectively (Skjemstad et al., 2004). The decomposition rate constants for the original formulation of the model are reported by Jenkinson and Rayner (1977). The decomposition of each active pool is assumed to increase, following first-order kinetics, with air temperature, but reduced by soil water deficits and the presence of vegetated soil cover. Temperature effects on soil

organic matter decomposition increase following a sigmoid function, while the topsoil moisture deficit reduces the effect by a factor of 0.2 to 1 (no moisture stress). The soil cover factor is 1.0 for bare soil and 0.6 when soil is vegetated, to slow organic matter decomposition. The main conceptual pools RPM, HUM and IOM are replaced with the measured particulate, mineral-associated (also referred to as humus in other research, but essentially measurements of organic C in the fine fraction with particle sizes $\leq 50\ \mu\mathrm{m}$), and resistant organic C fractions (POC, MAOC and ROC, respectively) (Skjemstad et al., 2004).

The POC fraction includes any DPM available in the soil at the time of measurement. The BIO pool was initially set to zero (Sparling, 1992).

## 2.2 Standardised soil C simulations

We simulated soil C dynamics across Australia (Figure 1), using a framework that enabled us to efficiently standardise and then integrate measurements and publicly available data on soil properties and environmental controls with ROTH C for simulation

at the scale of interest. The approach encompasses five stages as follows (Figure 1): 1) data compilation and synthesis, 2) data pre-processing and standardisation, 3) configuration of data on management regimes, and 4) model simulation supported by consistent initialisation and verification and 5) prediction.

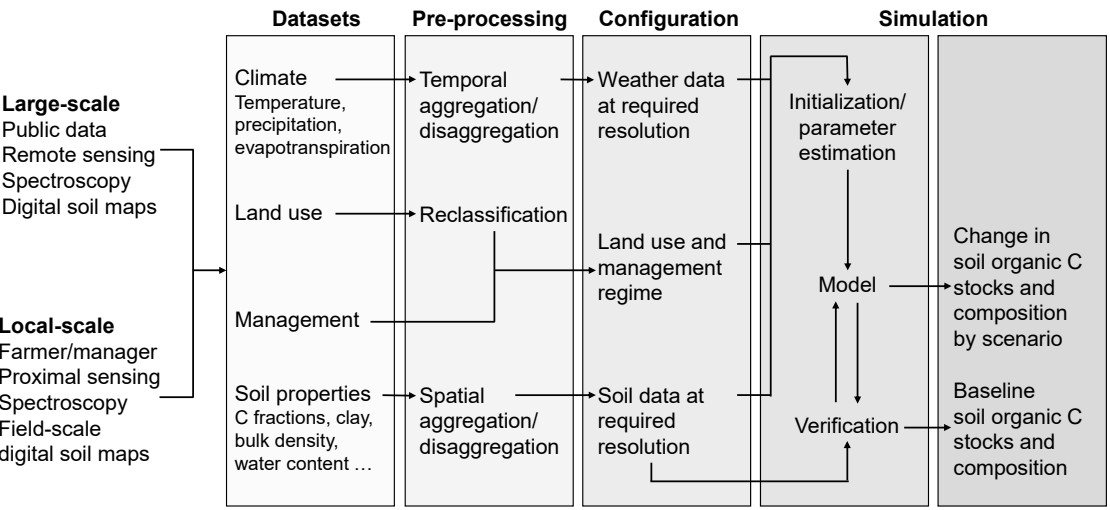

**Figure 1.** Soil carbon (C) simulation under a framework enables explicit standardisation and better connection between datasets and a soil process model at the appropriate scale.

## 2.3 Soil C simulations

### 2.3.1 Data compilation and synthesis

Roth C requires POC, MAOC, ROC, clay content, and sampling depth (in our case 0–0.3 m). The available water capacity (AWC) of the soil to a soil depth of 1 m is needed to calculate evapotranspiration from pan evaporation when a plant is present and to run a crop model (see below). We selected a total of 4,431 out of 5,721 sites across Australia (Viscarra Rossel et al., 2019) (Figure 2). The selected sites were under one of the dominant land uses: cropping, grazing of modified pastures and native vegetation, and natural conservation and protected areas (which includes deserts). Forests and production forestry

were excluded because we lack adequate data to support simulations under these land uses. The C fractions, clay content, and AWC were estimated with visible–near infrared spectra (Viscarra Rossel and Webster, 2012; Viscarra Rossel et al., 2015). Maximum air temperature, minimum air temperature, precipitation and pan evaporation are also required to run the model. We obtained gridded daily climate data (approximately 5-km resolution) from the SILO database of Australian climate data (SILO, 2020). We used the Australian Bureau of Agricultural and Resource Economics and Sciences land use map (ABARES,

2016) to determine detailed land cover across Australia. Agricultural activity data from Unkovich et al. (2017) provided data for croplands and modified pastures at Statistical Area Level 2 (SA2) (ABS, 2016), which are functional areas that represent socially and economically coherent communities. Current and historical events and agricultural practices, such as crop type and harvest, can be specified from 1970 to 2014. The other data required to run the model include an estimate of the decomposability of incoming biomass, soil cover, and monthly inputs of plant C and farmyard manure. These C input variables, if not measured,
must be estimated at each site (see below).

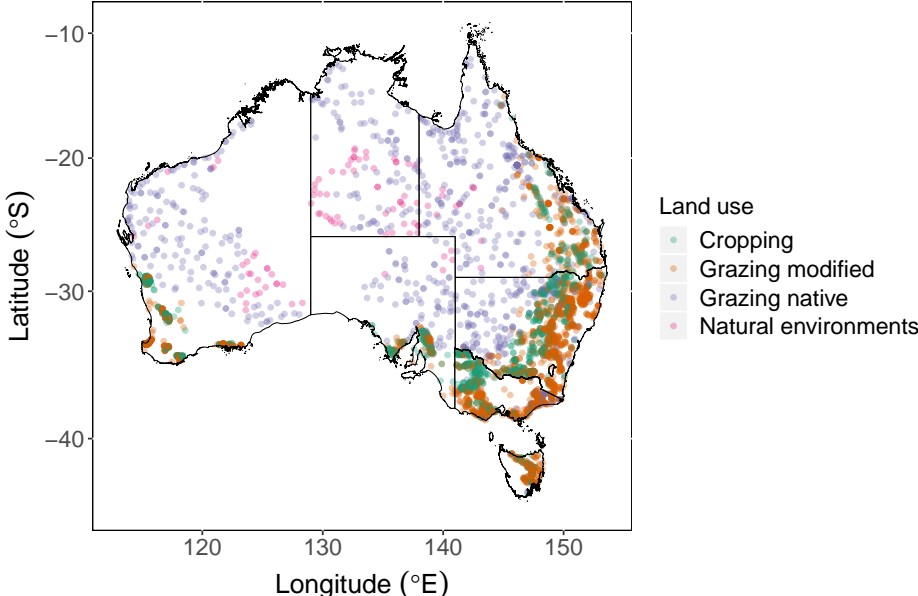

**Figure 2.** Location of 4,431 sites across Australia selected for this study. 1,261 sites were under cropping, 2,269 sites under grazing modified pastures, 807 sites under grazing of native vegetation, and 94 sites in natural environments, mostly under minimal use or managed resource protection in semi-arid and arid climates.

### 2.3.2 Data pre-processing and standardisation

The datasets were pre-processed and configured to provide consistent values and units of measurement. Daily weather was extracted at each of the 4,431 sites for the 20 years from 1991 to 2010. The mean of the minimum and maximum daily temperatures derived the average daily temperatures. Aggregation of the daily weather data produced monthly average temperature,
precipitation, and pan evaporation.

We used the Australian land use map (ABARES, 2016) to re-classify each site into the following broad land uses: cropping, modified grazing, native (unmodified) grazing, and natural environments. We defined cropping as land under broadacre crops. Modified grazing was defined as land used for livestock grazing on improved pastures with exotic vegetation cover. Native grazing was defined as land used for grazing on native pastures. Natural environments include the areas for nature conservation,

indigenous uses, and other minimal uses but exclude woodland and forest. We used the gridded Köppen climate classification from the Bureau of Meteorology (BOM, 2016) to identify sites under natural environments in semi-arid and arid climates. The area of cropping, modified grazing, native grazing, and natural environments occupy 292,104 km$^2$ (or 3.8%), 706,099 km$^2$ (9.2%), 3,439,468 km$^2$ (44.8%), and 1,507,616 km$^2$ (19.6%) of Australia, respectively. Data on agricultural practices at the SA2 level obtained from Unkovich et al. (2017) were used to select a crop or grass to represent typical management regimes in the sites under cropping and modified grazing.

### 2.3.3 Configuration of land management regimes and initial estimation of C inputs to soil

ROTH C does not calculate plant growth or the quantity of soil C inputs. Therefore, we estimated monthly plant C returns and farmyard manure added to the soil (e.g. managed or deposited by animals grazing on pasture) using the following approach. The initial estimate was made to set the starting values of the C inputs and to match the timing of C inputs to the crop or grass grown.

We assumed that crops were grown in rotations, but at sites under modified pastures, only a single grass species was considered. We used the activity data from Unkovich et al. (2017) to determine crop rotations and a representative grass species for each site during the baseline period between 1991 through 2010. For each of the periods 1990–1994, 1995–1999, 2000–2004, 2005–2009, 2010–2014, we calculated the cumulative frequency by regime. We used it to randomly select the crop or grass species (both annual and perennial) through time with a probability approach. The probability to have a certain crop was dependent on the cumulative frequency assigned to each crop type and regime. The crops grown in all years were selected and then used to determine the most dominant crop species. For the sites under native grazing, we considered a native perennial grass only.

For annual plant species, we used a crop model (Unkovich et al., 2018) that uses the amount of water available to the plant (derived from the measured AWC) to calculate a potential dry matter increment that is water-limited (WLDM) in kg ha$^{-1}$:

$$\text{WLDM} = ((ET \times Ts) + (DD \times Td)) \times TE$$

where $ET$ is the evapotranspiration (mm) from pan evaporation, $DD$ is any deep water drainage (mm) that occurs during the fallow season, $Ts$ is a fraction of ET that goes through the transpiration, $Td$ is a fraction of deep water drainage that goes through the transpiration, and $TE$ is the transpiration efficiency that is the amount of biomass produced per unit of water transpired (kg mm$^{-1}$) of a cropping or grazing system. Daily evapotranspiration was estimated by multiplying pan evaporation with a ratio of soil water content over plant AWC and the maximum evapotranspiration by crop or grass. The maximum dry matter production ($DM_{\max}$) is the sum of dry matter increments over the growing season. This model then back calculates dry matter accumulation (kg ha$^{-1}$) over the season ($DM_{\text{acc}}$):

$$\text{DM}_{\text{acc}} = \frac{DM_{\max}}{\left(1 + e^{-\frac{Day - a \times Days_{\max}}{b \times Day_{\text{sow}} \times Days_{\max}}}\right)}$$

where $Day$ is the current day as the season progresses, $Days_{\max}$ is the number of total growing days, $Day_{\text{sow}}$ is the day of planting, and $a$ and $b$ are growth coefficients specific for the plant. For a perennial system, daily growth (G) in kg ha$^{-1}$ is

calculated as:

$$G = WLT \times TI \times TE$$

where $WLT$ is the amount of water-limited transpiration (mm) that is evapotranspiration multiplied by vegetation cover, $TI$ is the temperature index function (Nix, 1981), and $TE$ is the transpiration efficiency of a perennial system (kg mm$^{-1}$). The perennial plant growth is used to calculate dry matter accumulation over the season. The model estimates root biomass using a fixed root-to-shoot ratio of 0.3 (Bolinder et al., 1997).

For both modified and native pastures, we assumed grazing to occur if the grass accumulated 1.2 Mg ha$^{-1}$ of shoot dry-matter, with no grazing effect on its growth (DPIRD, 2020). The start of grazing was set to optimise the pasture growth throughout the year, based on recommendations for efficient green pasture utilisation in Australia (MLA, 2019). We also assumed that grazing animals consumed 50% of daily shoot growth, returned 50% of the consumption to the soil as dung, and shed 50% of daily root growth. When the available soil water fell to $< 15\%$ of water holding capacity, 1% and 0.5% of the shoot dry matter and the root dry matter were assumed to die daily. The C content of above-ground and below-ground residues was 42% by mass, which is the value used in the FullCAM (Richards and Evans, 2004).

For the sites under natural environments, however, we did not use the plant model because we had no data on plants in this region. Instead, we assumed small but consistent C inputs from plant residues only (Wang and Barrett, 2003), which we set to 0.049 Mg ha$^{-1}$/month. No soil cover was assumed because in these regions, vegetation cover is typically sparse.

### 2.3.4  Simulation: optimisation of C inputs to the baseline soil organic C

We initialised the stocks of POC, MAOC and ROC pools using the measured data at 4,431 sites. We assumed that the initial soil organic C stocks were at equilibrium, and ran the model to reproduce their equilibrium condition. We based our assumption on data from the National Carbon Accounting System (NCAS) that include temporal soil organic C changes at 73 sites in Australia, recorded from 1911 to 2000 (Skjemstad and Spouncer, 2003). The DPM/RPM ratio determines the decomposability of incoming biomass. By default, the recommended DPM/RPM ratio is 1.44 for most crops and improved pastures and 0.67 for unimproved grasslands (Coleman and Jenkinson, 1996). The DPM/RPM ratio depends on the quality of C in plant residues and manure. It is site-specific, differs with land-use (Post and Kwon, 2000), and is unknown for Australian native grazing or natural environments (including deserts).

We tested six different DPM/RPM ratios (0.67, 0.96, 1.17, 1.44, 1.78 and 2.23) to estimate baseline C inputs and to assess the sensitivity of the simulated TOC, POC and MAOC to this parameter. These chosen ratios correspond to allocations of incoming plant material to DPM in the range 40–69%, and proportionally, to RPM in the range 60–31%. For each DPM/RPM ratio, we run the simulations at each of the 4,431 sites for 100 years. Specifically, for each ratio at each location, we performed the simulations iteratively up to 1000 times (or less if the model achieved equilibrium), by re-initialising the POC and MAOC pools with the measured C fractions and with a change in monthly input of plant residues and farmyard manure equivalent to 1/100 of their initial values. We considered only monthly C inputs in the simulations. The weather data used in the simulations represents the conditions of the baseline period between 1991–2010, which were repeated over the 100-year period.

Equilibrium condition occurred when 1) both POC and MAOC did not significantly change over time ($P > 0.05$), or 2) we observed an absolute change of $< 0.0025$ Mg C ha$^{-1}$ in both POC and MAOC. We used a time series linear model with a trend and seasonality to fit the change in POC and MAOC over time. An equilibrium condition was also assumed if the direction of the trend (positive or negative) in either pool changed. This condition prevented unrealistic simulations because both POC and MAOC showed the same trend in response to C inputs. Depending on the DPM/RPM, at 12 to 14 out of the 4,431 sites, the model was not able to simulate the equilibrium condition. We note that, for the sites that failed, changing C inputs only is insufficient for making both the POC and MAOC pools reach equilibrium simultaneously.

We report the stocks of TOC, POC and MAOC at the end of the 100-year simulation. The difference between the measured and the simulated TOC stock provided an estimate of the model deviation. We also calculated the range of monthly variation in simulated TOC stocks. For each site, we selected the DPM/RPM ratio based on the minimum deviation of TOC. Three hundred and eighty-eight sites had a model deviation and range of monthly change in TOC stock $\geq 10$ Mg C ha$^{-1}$, so we excluded them. We based the 10 Mg C ha$^{-1}$ threshold on the range of measured annual changes in TOC. The median TOC stock at these sites was 75.04 Mg C ha$^{-1}$ (range 52.58–111.44 Mg C ha$^{-1}$), and mostly, they occurred under modified grazing (data not shown). Finally, we optimised the amount of monthly C input and the DPM/RPM ratio at 4,043 sites and used them as the baseline. We determined the dominant values of the DPM/RPM ratio for each land-use across Australia, based on their relative frequency.

### 2.3.5 Simulation: the potential for C sequestration under changing C inputs

Using the calibrated model, we simulated potential changes in soil organic C over 100 years, in response to changes in C inputs. We selected different rates of C input to the soil by multiplying the optimised baseline with the factors 0 (no input), 0.25, 0.5, 0.75, 1.25, 1.5, and 2. These rates were selected to represent a wide range of C input levels that would be either physically achievable or manageable (e.g. manure addition) (Maillard and Angers, 2014). The increase in C input was restricted to a maximum of two times the baseline C inputs. Scenarios that varied the timing or quality of C inputs were not considered because we already calculated the timing of C inputs and the sensitivity to the DPM/RPM ratio. We chose 100 years in order to simulate the long-term response of TOC, POC and MAOC and calculated 11-year moving averages of the stocks of TOC, POC and MAOC, and the potential vulnerability of soil C to decomposition (POC/(MAOC + ROC), (Viscarra Rossel et al., 2019)). We calculated changes in soil organic C by changing C inputs and report the median stocks and lower and upper 95% confidence intervals (Conover, 1998) for the last 11 years of the simulation, when it reached a new equilibrium.

### 2.3.6 Simulation: the potential for C sequestration under a changing climate

We simulated the potential changes in soil organic C in response to projected changes in climate, and using the estimated baseline C inputs (as described in section 2.3.4). To do this, the baseline weather data was modified by adding temperature increases of 1.5°C, 2°C and 5°C, respectively, which fall within the likely range of mean annual temperature change from the Coupled Model Inter-comparison Projects sixth assessment report (CMIP6) (Tebaldi et al., 2021). To account for the CMIP6 projected changes in precipitation (Tebaldi et al., 2021), we also used changes of -5 %, -10 % and -15 %, respectively. As the

projections of pan evaporation are also needed to run the model, we calculated these using the Hargreaves approach combined with Class-A pan coefficients (Hargreaves and Samani, 1982). Changes in soil organic C by changing climate were calculated and the median stocks and the first and third quartiles were reported for the last 11 years of the simulation.

### 2.3.7   Empirical assessment of controls on the simulated C change

There are soil and environmental controls on organic C that are not accounted for by ROTH C. To gain a better understanding of the controls on the change in soil organic C under changing C inputs, we modelled the change in TOC, POC and MAOC as a function of the four land-use classes and a set of environmental variables. The environmental variables included i) soil properties, such as total nitrogen (N), total phosphorous (P), and C:N (Viscarra Rossel et al., 2015), ii) climate, iii) clay minerals (illite, kaolinite, and smectite) (Viscarra Rossel, 2011) and iv) potassium (K), thorium (Th) and uranium (U) from gamma radiometrics, which represent mineralogy and parent material (Minty et al., 2009). For the modelling, we used the machine learning method CUBIST (Quinlan, 1992). Briefly, CUBIST uses a recursive partitioning of the predictor variable space and divide the data into subsets that are more similar with respect to the predictors in the data (Quinlan, 1992). A series of rules derived from if-then conditions define the partitions, and each condition is based on a threshold for one or more of the predictors. When the conditions in each rule are satisfied, piecewise linear least squares regressions are used to model the response within each partition. To build precise and stable models, we tested combinations of committees (1, 2, 5, 10, and 20) and the number of neighbours (0, 2, 5, and 9) using 10-repeated cross-validation (Hastie et al., 2009). We used the minimum root mean squared error (RMSE) to select the best model. We then assessed the relative importance of each variable based on the usage of each variable in the rule conditions and the models for Cubist.

## 3   Results

### 3.1   Effect of different quality of C inputs on soil organic C

The median stocks of TOC, POC and MAOC in the 0–0.3 m soil layer, calculated across Australia are 26.01, 3.43 and 16.10 Mg C ha$^{-1}$, respectively (Figure 3). The measured TOC stocks under natural environments, native grazing, modified grazing and cropping are 15.45 Mg C ha$^{-1}$ (interquartile range 11.89–18.11 Mg C ha$^{-1}$), 24.61 Mg C ha$^{-1}$ (18.96–34.17), 51.48 Mg C ha$^{-1}$ (39.01–74.60) and 35.38 Mg C ha$^{-1}$ (25.39–43.55), respectively. The POC and MAOC fractions consist of 11% and 68% of the measured TOC stocks under natural environments (including deserts), 11% and 67% under native grazing, 18% and 52% under modified grazing, and 16% and 53% under cropping.

   With each of the DPM/RPM ratios tested, the model simulated the measured TOC, POC and MAOC stocks at equilibrium (Supplement Table S1), but the amount of annual C input needed to maintain the soil organic C stocks was sensitive to the varying quality of incoming plant material. The C inputs increased from 1.47 to 1.83 Mg C ha$^{-1}$ when the DPM/RPM ratio increased from 0.67 (low decomposability) to 2.23 (high decomposability). With those changes, the rate of C inputs into DPM

rose from 0.59 to 1.26 Mg C ha$^{-1}$ yr$^{-1}$, while the rate into RPM decreased from 0.88 to 0.57 Mg C ha$^{-1}$ yr$^{-1}$. The addition of biomass C with different qualities affected the levels of POC and MAOC at equilibrium (Supplement Table S1).

With an optimised DPM/RPM ratio at each location, the model was able to explain 97–98% of the measured variation in TOC at sites under native grazing, modified grazing and cropping. RMSE values ranged from 2.45 to 3.55 Mg C ha$^{-1}$ (Figure 3). At locations under natural environments, the model explained only 65% of the variation in TOC, but with a similar RMSE of 3.22 Mg C ha$^{-1}$. The model could explain less of the variation in POC (55–89%) compared to TOC and MAOC. Also, the model did not perform as well for POC in the soils under cropping. Across Australia, the most frequent DPM/RPM ratio was

2.23 (1,773 sites), followed by the value 0.67 (for 829 sites) and 0.96 (415 sites) (Figure 3).

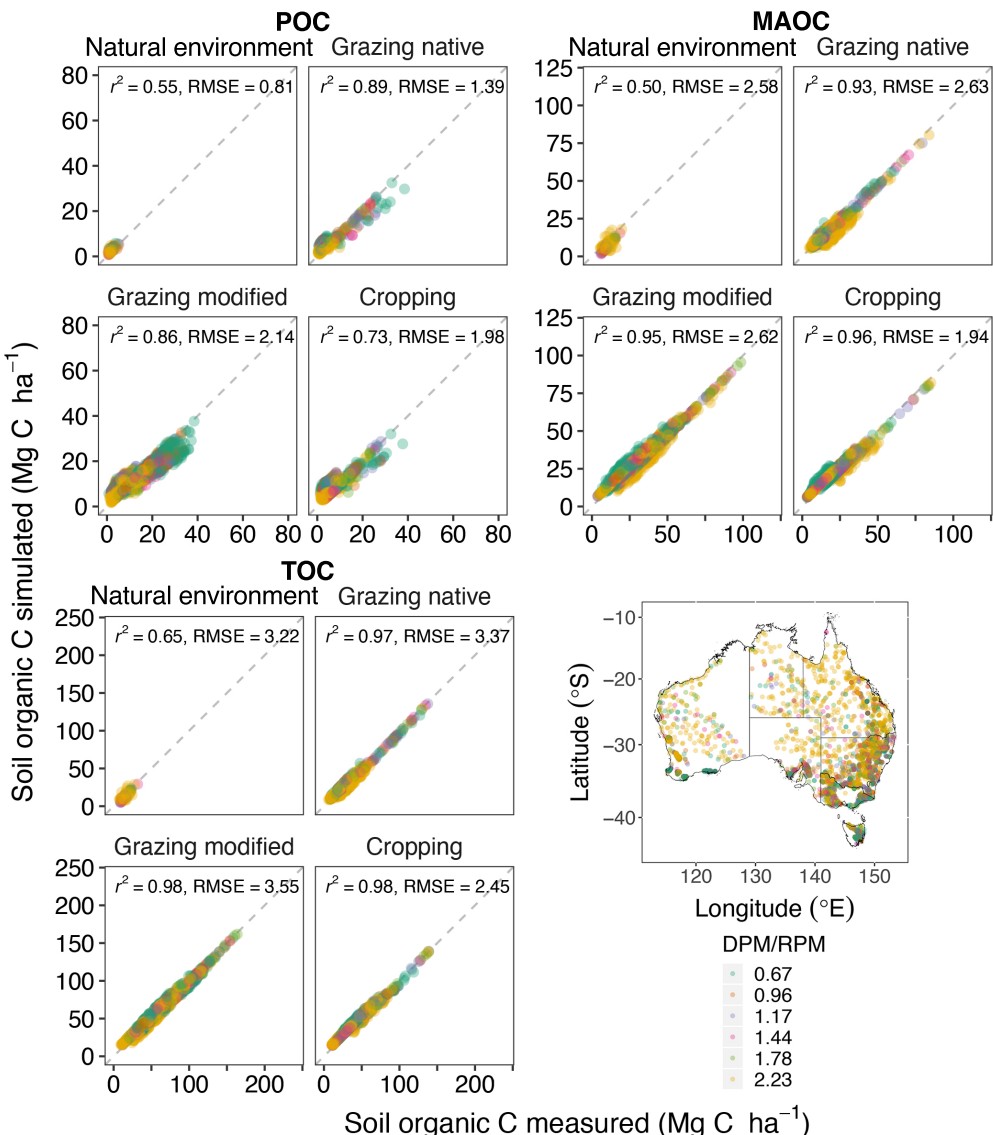

**Figure 3.** Simulation of equilibrium soil organic C levels after optimisation, based on its sensitivity to the changes in the allocation of incoming plant and manure C into the decomposable plant material (DPM) and resistant plant material (RPM) components (n = 4,043). The map shows the geographic distribution of DPM/RPM values, where high values correspond to faster decomposition.

The simulated median and total TOC, POC and MAOC stocks for each land-use class and overall were similar to the measured data (Table 1).

**Table 1.** The simulated stocks of total, particulate and mineral-associated organic C (TOC, POC and MAOC) under different land uses and in Australia. The total stocks of soil organic C were calculated from the median with the uncertainties expressed as an approximate 95% confidence interval (CI).

| Land use | | Median (Mg/ha) | 1st quantile (Mg/ha) | 3rd quantile (Mg/ha) | Total (Gt) | Lower 95% CI (Gt) | Upper 95% CI (Gt) |
|---|---|---|---|---|---|---|---|
| Cropping | TOC | 40.01 | 29.71 | 46.43 | 1.18 | 0.87 | 1.15 |
| (n = 1182) | POC | 7.68 | 6.30 | 9.23 | 0.22 | 0.18 | 0.27 |
| | MAOC | 21.91 | 16.51 | 25.43 | 0.64 | 0.48 | 0.74 |
| Grazing modified | TOC | 51.84 | 42.77 | 67.58 | 3.74 | 3.02 | 4.77 |
| (n = 2008) | POC | 9.93 | 7.86 | 13.04 | 0.70 | 0.55 | 0.92 |
| | MAOC | 27.29 | 22.91 | 35.34 | 1.93 | 1.62 | 2.50 |
| Grazing native | TOC | 23.15 | 18.00 | 32.34 | 8.12 | 6.19 | 11.12 |
| (n = 777) | POC | 3.71 | 2.65 | 5.60 | 1.28 | 0.91 | 1.93 |
| | MAOC | 13.59 | 10.83 | 18.14 | 4.67 | 3.73 | 6.24 |
| Natural environments | TOC | 12.40 | 10.21 | 17.54 | 2.12 | 1.54 | 2.64 |
| (n = 76) | POC | 2.09 | 1.63 | 3.04 | 0.31 | 0.25 | 0.46 |
| | MAOC | 7.29 | 5.53 | 10.27 | 1.10 | 0.83 | 1.55 |
| Australia* | TOC | 25.45 | 19.37 | 34.94 | 19.82 | 14.63 | 24.52 |
| (n = 4043) | POC | 4.46 | 3.23 | 6.44 | 3.43 | 2.24 | 4.64 |
| | MAOC | 14.22 | 10.75 | 19.10 | 10.91 | 8.37 | 12.52 |

* The Australian-wide estimates were the area weighted averages of the medians for the four land-use classes. The area of cropping, modified grazing, native grazing, and natural environments (including deserts) occupy 3.8%, 9.2%, 44.8%, and 19.6% of Australia (total area 7673138 km$^2$), respectively.

## 3.2 Effect of changing C inputs on soil organic C

The TOC, POC and MAOC stocks at equilibrium were positively related to the level of C inputs (Figure 4). Annual C inputs to the soil under natural environments, native grazing, modified grazing and cropping were 2.38, 0.77, 1.86 and 1.60 Mg C ha$^{-1}$, respectively. Therefore, the model estimated the largest amount of C inputs required to maintain soil organic C under natural environments compared to the other land uses. The corresponding interquartile range was 1.11–3.57 Mg C ha$^{-1}$ for natural environments. In comparison, there was a wider range of C inputs for native grazing (0.57–1.13 Mg C ha$^{-1}$), modified grazing (1.37–3.01 Mg C ha$^{-1}$), and cropping (1.20–2.18 Mg C ha$^{-1}$) (Figure 4). For the agricultural soils, clay affected the relationship between soil organic C stocks and C inputs as soil with more clay (predominantly in eastern Australia) could hold more

organic C (Figure 4). Under grazing and cropping land uses, the response of MAOC and TOC to increasing C inputs appears to depend on clay content. This pattern was not evident for POC as this pool is not directly associated with clay in the model.

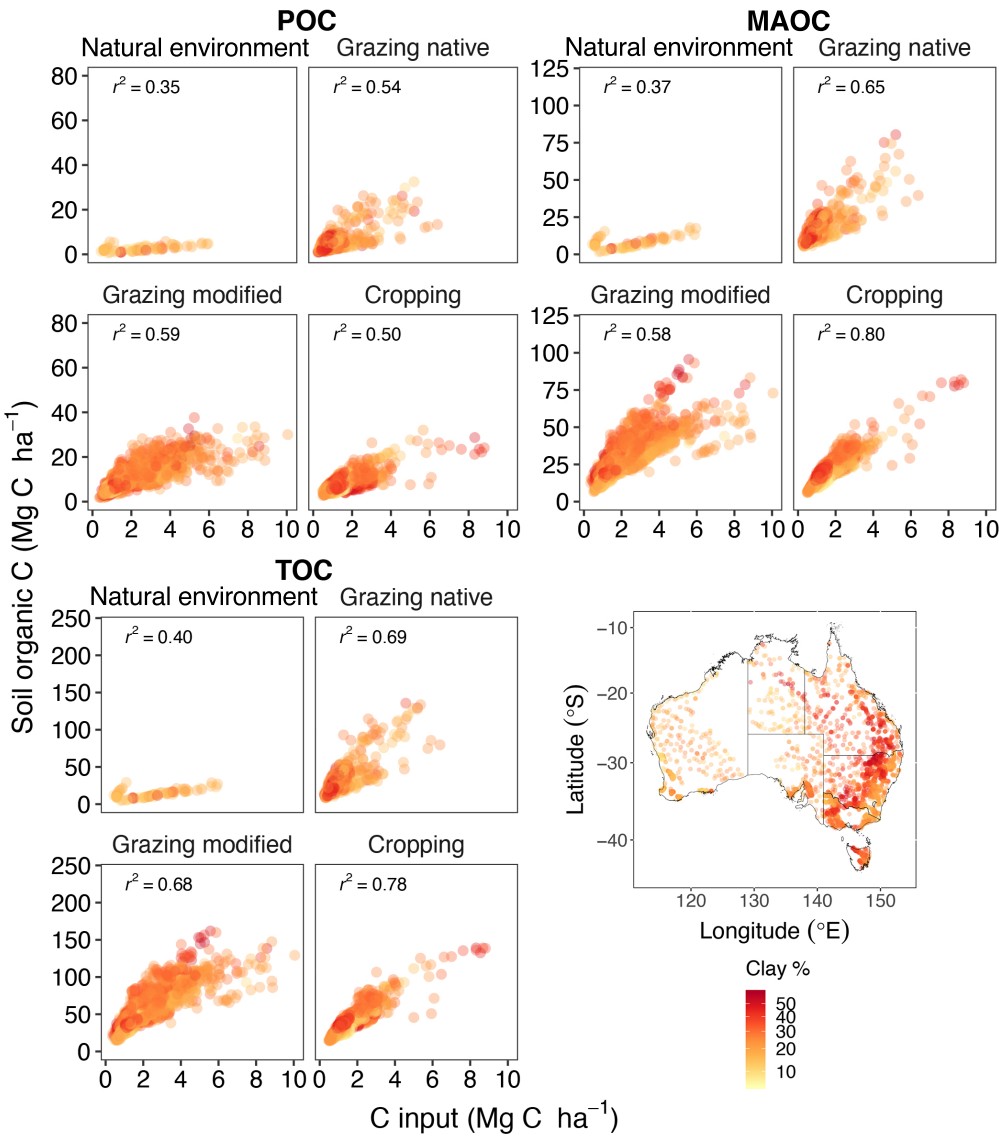

**Figure 4.** Changes in total, particulate and mineral-associated organic C (TOC, POC and MAOC) with C inputs by land use (n = 4,043).

The model explained 78, 80, and 50% of the variation in TOC, MAOC, and POC by increasing C input under cropping (Figure 4). The relationship was poorer under native and modified grazing ($r^2$ = 0.54–0.69) (Figure 4). There was a relatively weak and divergent relationship between soil organic C stocks and C inputs to the soil under natural environments ($r^2$ = 0.35–

0.40), mostly due to differences in precipitation. We found that soil organic C was more responsive to C inputs at the sites with less annual precipitation (approximately 170 mm).

After 100 year simulation, the TOC, POC and MAOC stocks (at or near a new equilibrium) responded linearly to changing soil C inputs from the baseline (Figure 5a). The median changes in POC and MAOC over the last 11 years of simulation (Figure 5b) show that with increasing C inputs, the soil under native grazing, modified grazing and cropping, respectively, were the most potentially vulnerable to C loss (Viscarra Rossel et al., 2019)because in these soils, there was a larger proportional increase in POC relative to MAOC. Soil under natural environments was the least vulnerable to C loss because of the smaller increase in POC relative to MAOC.

With an annual increase of 1 Mg C ha$^{-1}$ in C inputs from the baseline and under current climatic conditions, soils under natural environments can potentially increase TOC stocks by 3.52 Mg C ha$^{-1}$. In this case, the stocks of POC and MAOC increased by 0.92 Mg C ha$^{-1}$ and 2.48 Mg C ha$^{-1}$, respectively. Soils under the other land use were more sensitive to increasing C inputs, when added to soil with less intensive management. Under native grazing, TOC, POC and MAOC stocks changed by 15.57 Mg C ha$^{-1}$, 5.49 Mg C ha$^{-1}$ and 9.18 Mg C ha$^{-1}$, respectively, and at the same rate as the C inputs in the simulation. Under modified grazing, TOC stocks changed by 14.21 Mg C ha$^{-1}$ and POC and MAOC accounted for 5.34 Mg C ha$^{-1}$ and 8.12 Mg C ha$^{-1}$ of the change, respectively (Figure 5b). Changes in TOC, POC and MAOC stocks under cropping were 13.58 Mg C ha$^{-1}$, 4.69 Mg C ha$^{-1}$ and 8.35 Mg C ha$^{-1}$, respectively. When C inputs decreased, POC and MAOC were depleted in native grazing systems at a rate about two times greater than the other land use types.

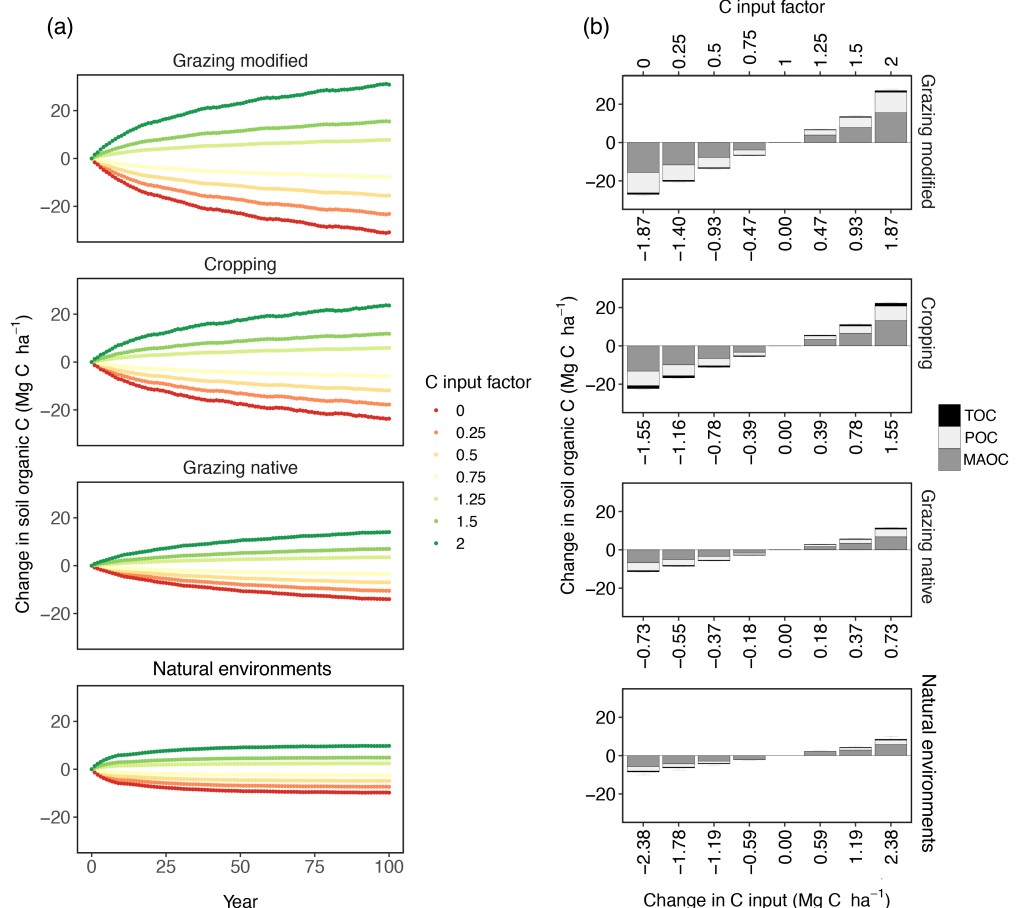

**Figure 5.** (a) The 100-year simulations showing the changes in total organic C (TOC) in the topsoil (0–0.3 m) following changes in C input (n = 4,043). At each site, baseline C input was multiplied by the factor 0, 0.25, 0.5, 0.75, 1.25, 1.5, and 2 to derive different C input levels. (b) Median changes in TOC and its fractions, consisting of the particulate and mineral-associated organic C (POC and MAOC), calculated over the last 11 years of the simulation (at the new or near new equilibrium).

## 3.3 Effect of changing climate on soil organic C

By the end of the 100-year simulation, the median TOC stocks under natural environments decreased by 3.1–4.5 Mg C ha$^{-1}$
with temperature increases of 1.5–5°C, respectively (Figure 6). This loss corresponds to around 21–32 % of the baseline stocks, and the proportion of POC and MAOC lost was similar (Table 2).

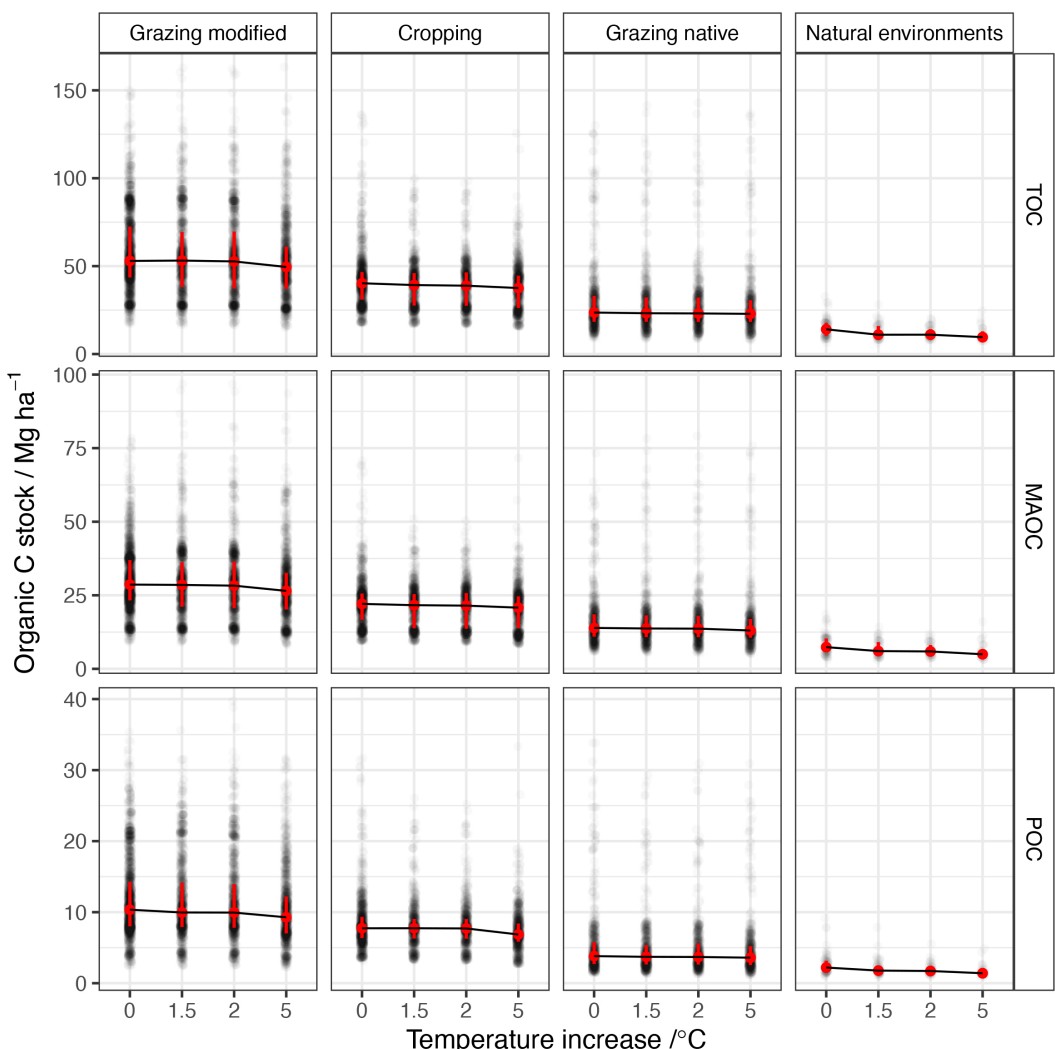

**Figure 6.** After 100-year simulations, differences in total organic C stock under a warming climate, showing median values and the 10th and 90th quantiles. A changing climate is represented by adding 1.5°C, 2°C and 5°C to the baseline temperatures (1991–2010) and then repeating over the 100-year period.

Soil under cropping lost between 1.0 and 2.7 Mg C ha$^{-1}$ with temperature increases of 1.5–5°C, respectively (Figure 6), corresponding to around 2.5–6.8 % of the baseline stocks (Table 2). With temperature increases of 1.5°C and 2°C soil under cropping lost more MAOC than POC, however, with a 5°C change, the loss of POC was greater than the loss of MAOC (Table 2). The loss of TOC under native grazing ranged from 0.4 to 0.7 Mg C ha$^{-1}$ with temperature increases of 1.5–5°C, respectively (Figure 6). The loss of POC was larger than the loss of MAOC with temperature increases of 1.5°C and 2°C, however with a temperature increase of 5°C the loss of both POC and MAOC was large and proportionally similar (Table 2). Under modified grazing, warming by 1.5°C produced and increase in TOC of 0.2 Mg C ha$^{-1}$, but with further increases of 2

and 5°C, the median loss of TOC was 0.28 and 3.4 Mg C ha$^{-1}$, respectively. The proportion of POC lost was greater than the
loss of MAOC (Table 2).

**Table 2.** After 100-year simulations, median change (%) in the stocks of total, particulate and mineral-associated organic C (TOC, POC and MAOC) from baseline as a result of climate change, represented by temperature increases of 1.5°C, 2°C and 5°C.

|  | Temperature increase (°C) | Cropping | Grazing modified | Grazing native | Natural environments |
|---|---|---|---|---|---|
|  | 1.5 | -2.54 | 0.38 | -1.67 | -21.87 |
| TOC | 2 | -3.44 | -0.53 | -2.13 | -21.66 |
|  | 5 | -6.75 | -6.46 | -3.12 | -31.65 |
|  | 1.5 | -0.06 | -3.78 | -2.62 | -19.51 |
| POC | 2 | -0.34 | -3.94 | -3.26 | -22.03 |
|  | 5 | -11.29 | -10.32 | -5.74 | -36.35 |
|  | 1.5 | -1.98 | -0.42 | -1.33 | -18.28 |
| MAOC | 2 | -2.69 | -1.28 | -1.78 | -19.94 |
|  | 5 | -5.79 | -7.50 | -6.22 | -32.79 |

## 3.4 The controls on the simulated soil organic C change

Climatic variables, particularly temperature and potential evaporation, controlled the changes in TOC, POC and MAOC (Figure 7). Clay content had a dominant effect on the changes in MAOC because in ROTH C, clay determines the ratio of $CO_2$ released to MAOC formed, during decomposition. Total N, the C:N ratio and pH were important controls for the changes in POC (Figure 7), and might be related to a capacity of the soil to form POC. Cropping affected the changes in POC, possibly because of the crop-specific distribution of C inputs. The controls on POC were similar to those on TOC because their changes were proportional. The land use in natural environments affected the changes in MAOC (Figure 7), suggesting that we need a greater understanding of the potential for C sequestration in low clay content soils in hot and dry climates.

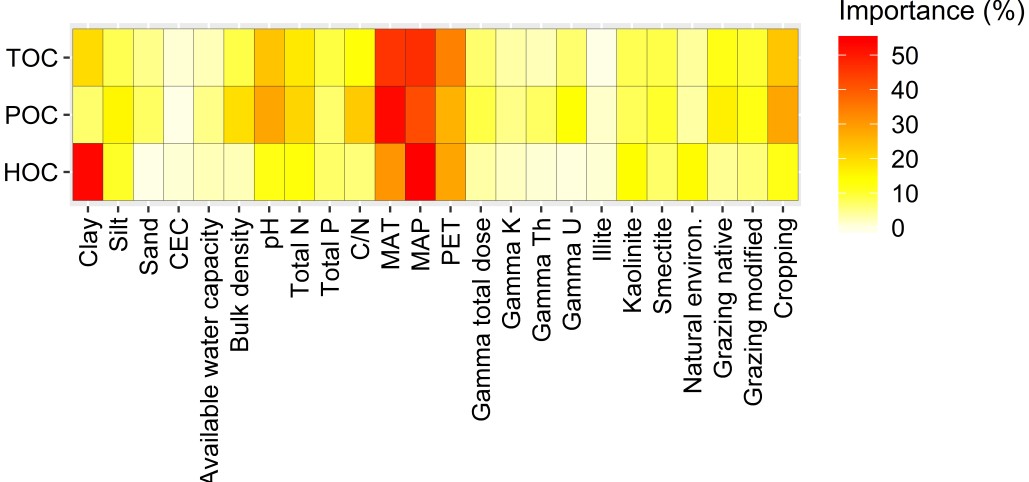

**Figure 7.** Importance of the environmental variables that contribute to potential changes in total, particulate and mineral-associated C organic C (TOC, POC and MAOC) by changing C inputs. Climatic variables, i.e. mean annual temperature (MAT), mean annual total precipitation (MAP) and potential evapotranspiration (PET), are averaged over a period of 1991–2010. CEC is the cation exchange capacity of a soil. The importance of each soil variable was assessed based on the usage of each individual variable in the rule conditions and the model for Cubist.

## 4    Discussion

### 4.1    Current estimates for soil organic C stocks in Australia

We used ROTH C because it requires few parameters, it initialises its main pools with measured C fractions, it was adjusted to suit Australian conditions (Janik et al., 2002; Skjemstad et al., 2004) and has been shown to perform well over a wide range of conditions world wide (Farina et al., 2013; Poeplau and Don, 2015). Further, the model is in Australia's National Greenhouse Gas Inventory System and the ERF, and so we thought it useful to comply. The climatic and soil property inputs needed to run ROTH C are readily available from publicly available datasets (see Methods) or are relatively easily measured, for instance, with proximal sensors (England and Viscarra Rossel, 2018).

The main soil C pools of ROTH C can be initialised with measured C fractions (POC, MAOC, ROC), there is no need for spin-up simulations (i.e. simulations until the model reaches equilibrium), making it possible to run the model site-specifically at any location in Australia. Further, using measured C fractions in the model allows for the assignment of the primary pool structure and the measurements serve as internal verification of the model. In our case, we empirically assessed how well the baseline simulations matched the model's corresponding dynamic pools, which suggest that the model is able to represent

Australian soils. Such data-driven model initialisation helps with the selection and site-specific estimation of 'unknown' model parameters, such as the amount and quality of C inputs, which is important for a more consistent calibration of the model (Aber, 1997; Seidel et al., 2018). Our simulations successfully optimised both the amount and the quality of C inputs to maintain the current baseline soil organic C stocks.

The model explained 73–98% of the variation in the size of the C pools in soils that are under cropping and 86–98% of that under grazing, while the simulation under natural environments in semi-arid and arid climates need improving. Together with the relatively large C inputs, required to maintain baseline TOC (9.61 to 17.05 Mg C ha$^{-1}$), this poor performance suggests that the model did not represent well the complex decomposition processes described by the (hot and dry) climate and soil under natural environments. We hope to address this in subsequent research because soil C in semi-arid and arid climates might represent a crucial C sink in Australia and other similar regions of the world (Farina et al., 2013).

The simulated baseline estimate of the total TOC stock in Australia is 19.52 Gt, which is less than the 24.97 Gt estimate of Viscarra Rossel et al. (2014), as soils under land uses that contain more carbon, e.g. forests were not included in this study. Our estimates for soil under natural environments and native grazing are 1.96 Gt (with 95% confidence intervals of 1.65–2.30 Gt) and 8.02 Gt TOC (7.90–8.54 Gt), respectively. The soil under native grazing has the largest organic C stocks compared to the other land uses. The contribution of native grazing to the national soil organic C budget is considerable due to the large extent of land that it covers. This estimate was well within the confidence intervals derived by Viscarra Rossel et al. (2014), although slightly larger. Estimates of the total TOC stocks for soils under modified grazing and cropping are 3.79 Gt (with 95% confidence intervals 3.72–3.86 Gt) and 1.18 Gt (1.16–1.20 Gt), respectively. These estimates were also somewhat larger than those of Viscarra Rossel et al. (2014). Our estimates of the total POC and MAOC stocks across all four land uses are 3.43 and 10.91 Gt, which are smaller than the 7.8 and 27.3 Gt, respectively, estimated derived by Viscarra Rossel et al. (2019). However, our estimates are within the range of their confidence intervals. A reason for the differences between our estimates and those of Viscarra Rossel et al. (2014, 2019) might be that our estimates from the simulations are based on a relatively sparse sample (Figure 1), while theirs are from a complete enumeration of Australia with spatial machine learning models. Nevertheless, the results from our simulations suggest that the ROTH C model can explain the soil processes under different land uses tested, which are important for estimating the baseline total stocks of soil organic C and its composition.

## 4.2  Possible future change in the organic C stocks of Australian soils

There are few quantitative assessments of soil C dynamics in Australia. Primarily they are for cropping regions (Luo et al., 2014; Lam et al., 2013; Wang et al., 2016), some present local case studies (Hoyle et al., 2013), and some report estimates that are uncertain because of the lack of comprehensive surveys and scarcity in data (Gifford, 2010). Here, we simulated soil organic C at 4,043 sites across Australia to estimate changes in C stocks from a range of plausible changes in C inputs to the soil. With an annual increase of 1 Mg C ha$^{-1}$ in C inputs, the model estimated the largest potential soil C increase in soil under native grazing (12.07–17.82 Mg C ha$^{-1}$), followed by modified grazing (12.38–16.03 Mg C ha$^{-1}$). The potential increase in soils under cropping was smaller (12.19–15.80 Mg C ha$^{-1}$), possibly due to the effect of soil disturbances and cultivation on decomposition. However, the difference between grazing and cropping is small as the effects of climate and soil texture on

organic matter and its decomposition are likely to be similar over the large areas that these land uses occupy. Soils in natural environments had the smallest potential to accumulate C (3.15–4.09 Mg C ha$^{-1}$), because they occur over large areas with semi-arid to arid climates characterised by low precipitation, generally below 500 mm yr$^{-1}$ and high temperatures up to 50°C (ABS, 2016).

The simulations that account for climate change suggest that Australian soils will become more vulnerable to C loss. The changes in climate also mean that opportunities for managing TOC (and C composition) will be affected by the feedbacks on plant productivity and hence C inputs (Pareek et al., 2020; Paustian et al., 2019). Compared to the other land uses, soil that is under natural environments appeared to be the most sensitive to climate change, showing a potential decrease of 3.1–4.5 Mg C ha$^{-1}$ (or 22–32 % of their stock) with temperature increases of 1.5–5°C. This is significant because, although these

soils hold the smallest median C stocks, they hold a relatively large total stock (e.g. compared to cropping soils) because natural environments cover a large extent, and a larger proportion of the C is in the more stable, MAOC fraction (Table 1). The model predicted that these soils need a large amount of C inputs to maintain current C stocks, however, in these areas primary productivity is small and will be further limited by the predicted warmer and drier future climates (Haverd et al., 2016). Our results also show that soil that is under native grazing, which require less C inputs to maintain stocks than soil that is under

natural environments, are less sensitivity to the same temperature increases, with decreases in TOC of 0.4–0.7 Mg C ha$^{-1}$ or 1.7–3.1% of their total stock. In more managed systems, such as cropping and modified grazing, the potential loss of organic C in soil due to climate change will, to some extent, be compensated by management and additional C inputs into the soil. For example, our results indicate that in cropping systems with annual additions of 1 Mg C ha$^{-1}$ more than the baseline annual C input, temperature increases of 1.5°C, 2.0°C and 5°C will offset of around 6%, 9%, and 20%, respectively, of increased TOC

stocks. We note the need to understand better the mechanisms of C stabilisation and its interaction with climate change and to develop management strategies that store new C sequestered in these soils.

### 4.3   Carbon inputs

We did not use net primary productivity (NPP) as a proxy for C inputs to the soil. Although large-scale estimates of NPP might be a good proxy for the C inputs in natural environments, they would be inadequate for managed systems (Haverd et al.,

2013). To derive estimates of NPP for managed land uses, such as croplands, one needs fine spatial resolution land cover data with crop-specific information (Li et al., 2014; Turner et al., 2006). These are not readily available continentally. Large-scale (global, continental or regional) estimates of NPP, such as those available from coarser resolution remote sensing, would not be suitable for agricultural environments, also because depending on the method used to derive NPP, the estimates would be largely uncertain (Roxburgh et al., 2005; Ciais et al., 2010). Therefore, using NPP as an estimate of C inputs for all four land

uses would have made our simulations more uncertain. We thought it important to maintain a consistent approach for deriving the C inputs, so we used a wide but plausible range of values to represent the C inputs across the whole of Australia. The range of C inputs that we used are representative of values that might be expected from management practices that enhance rates of primary production and C input to the soil, including manure addition (Lal, 2016; Paustian et al., 2019). Our results suggest that the baseline rate of C inputs is site-specific, and managing its rate locally is needed to avoid soil C loss from land-use

change. Importantly, these estimates of C inputs are useful to locate soils where C capture is possible under limited availability of water resources and nutrients (Baldock et al., 2012).

The long-term changes in organic C are primarily determined by the C inputs into the soil, and the sensitivity of the change can be affected by local conditions. The results from the empirical modelling suggest that simulations might improve if we can modify the environmental effects on decomposition, separately for each of the pools. For example, clay content did not importantly affect the changes in POC, but it did affect the changes in MAOC, otherwise known as the mineral-associated carbon (MAOC) (Lavallee et al., 2020). In contrast, other studies have shown that clay has a direct effect on both C inputs and the C pools in Australian soils (Krull et al., 2003; Luo et al., 2017). Of course, this might be due to the inability of the model to simulate textural controls on POC. Total N and the C:N ratio contribute more to the changes in POC than in MAOC. POC appears to be also affected by pH and more under cropping. These results demonstrate the difficulty that ROTH C has in simulating the more labile POC dynamics and the need to represent such additional environmental factors to better explain TOC change.

## 4.4 Simulating soil C dynamics using a standardise approach

There is a functional disconnect between measurements, data and biogeochemical models (Blankinship et al., 2018), but by simulating under a framework, like we did here, we can bridge that disconnect. A framework provides a standardised and consistent approach for organising and processing input datasets from different sources, to facilitate calibration, verification, estimation and prediction at an appropriate scale and resolution, depending on the study. The input data may originate from field or laboratory measurements, remote sensing, digital soil maps or other data from various sources. Using a standardised approach, soil C simulations can be more versatile. They can be performed on points, areas or pixels, even when few, or no site-specific data are available. In the latter case, by using fine spatial resolution information (Viscarra Rossel et al., 2014, 2015, 2019), or like we have shown here, one can use measurements together with publicly available continental-scale datasets and processes them consistently for the simulations. When site-specific data are available, then under the framework, they are processed appropriately for the local simulations, as we have shown in Lee and Viscarra Rossel (2020).

Simulating soil organic C in a standardised manner also facilitates consistent pre-processing, quality checks and explicit definition of the simulation unit. This is important because often, datasets have a different formats and resolutions, which must be standardised and harmonised before running the simulation (Batjes et al., 2020). Datasets may need to be aggregated or disaggregated over space and time, depending on the data and the need. For example, if the need is to run the simulations over a large-scale and over grids, finer resolution data e.g. soil property data, will require aggregation to match the coarser resolution of the simulation unit. Similarly, re-classification of categorical data, e.g. land-use data, may be performed, like we have done here, to set the spatial extent of the simulations. We used the model ROTH C, however, by using our approach, one could accommodate other soil C models, with only small changes to the workflow (Figure 1). This versatility is essential for extending our theoretical understanding of C cycling and its response to human-induced and environmental change at appropriate scales (Grunwald et al., 2011; Metting et al., 2001). Of course, with other multi-pool C models, it will be important to explore further the initialisation requirements and the baseline state for the simulations. The reason is that each soil C pool

could be at a different state. Other models may also drive decomposition based on different assumptions, e.g. soil enzyme
kinetics or microbial growth (Smith et al., 2020).

## 4.5 Future needs

Plant biomass production and subsequent C inputs to the soil are critical determinants of the quantity of organic matter in soil C
models. The simulations that we presented estimated the potential of soil C sequestration in response to changing C inputs and
climate under the main land uses in Australia. However, we will need data on plant growth properties, seasonal biomass data,
and residue and grazing management, to better represent management practices under these land uses. Without such datasets,
it is difficult to verify the balance between C inputs and the stocks and composition of soil organic C under different land-use
and management combinations, except for a few cropping systems (Wang et al., 2016). For the soils under native grazing, we
need new research on the specific growing conditions of plants (e.g. nutrient availability) and how they affect the amount and
timing of C inputs.

The machine learning could identify some other factors that contribute to the changes in soil organic C and determine their
relative importance. Although there is no direct mechanistic understanding gained from those analyses, some of those variables
are important predictors of soil C change, and they might need accounting in future model development. In practice, statistical
modelling can be incorporated in the simulations to help identify the balance of C flows between the soil, plant, and atmosphere
at the scale of interest. However, research to combine mechanistic and statistical modelling is still at an early stage, and more
research is needed to connect data with models (O'Rourke et al., 2015; Vereecken et al., 2016), in a consistent manner and
across scales, for example, Viscarra Rossel et al. (2019). With new measurements and subsequently growing datasets, we expect
to identify new processes and controls from statistical modelling and to further account for these in a standardised modelling
approach.

## 5 Conclusion

Our results show that the site-specific initialisation of the C pools with measurements of the C fractions (POC, MAOC, ROC)
are essential for accurately representing baseline soil organic C stocks and composition under different land uses. The source
and scale of these C data and other inputs drive the overall simulation process of soil C dynamics. We showed that, with
a site-specific optimisation of the DPM/RPM ratio, the model could explain 97–98% of the variation in TOC under native
grazing, modified grazing and cropping, respectively, and 65% under natural environments. The 100-year simulations showed
that, with an annual increase of 1 Mg C ha$^{-1}$, and under constant climate, the potential for C sequestration in Australian soils
is smallest in soils under natural environments, larger under cropping and modified grazing, and the greatest in the soils under
native grazing. The simulations also show that the effects of climate change on C sequestration will be largest in soils under
natural environments and least in soils under native grazing. Our simulations of soil organic C across Australia with ROTH
C were performed under a standardised approach that establishes a much-needed connection between measurements, datasets

and models. It enabled consistent processing of measurements and datasets from different sources, and standardisation and configuration of the model for calibration, verification, estimation and prediction under global changes.

*Code availability.* The various scripts used for data processing and simulation are available from the corresponding author on reasonable request.

*Data availability.* The spatial datasets on climate, soil and land use are publicly available from repositories cited in section 2.3.1. Other
datasets are available from the corresponding author on reasonable request.

*Author contributions.* RVR formulated the research and with JL designed the simulations. JL performed the simulation and with RVR the data analysis. RVR and MZ performed the climate change simulations. RVR and JL wrote the manuscript with input from YW, ZL, MZ.

*Competing interests.* The authors declare that they have no conflict of interest.

*Acknowledgements.* This work was supported by funding from Curtin University and the Australian Government through the Australian
Research Council's Discovery Projects funding scheme [project DP210100420].

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
