# Peer review of "Assessing the response of soil carbon in Australia to changing inputs and climate using a consistent modelling framework"

_Biogeosciences, 2020_

## Referee Comment (RC1) · Anonymous Referee #2 · 31 Jul 2020

General comments

The authors developed soil C calculating system which connects spatial datasets on meteorology, soil, land use and land management with the RothC model. They calibrated the RothC and predicted changes in soil C for 100 years with different soil management scenarios. I think this work is within the scope of this journal and potentially many of audience of Journal would be interested in. This was my first impression after quick read of paper. But after careful reading, I found some severe problems. First, explanation of what the authors have done is not enough throughout the paper especially in "Materials and methods" section. Frequent disconnection in logic between

sentences made me difficult to understand what the authors really have done. Development of calculation system is great achievement, but it was difficult to evaluate the validity of the many assumptions in developing the system and future simulation procedure. This might be partly because of English skill but I think not only due to that. Significant re-writing of manuscript with English check by native speaker will be needed for this manuscript.

Second, future projection of 100 years generally requires the use of climate change scenarios but there is no description on this. I understood that the future projection in this study was conducted by using current meteorological data. This is curious.

Third, the setting of the changing amount of C input in future projection is not realistic. I do not think six times higher organic matter input is realistic scenario. It is natural that increasing C input result in higher soil C qualitatively of course. Quantitative estimation by using realistic scenario (both future climate change and management scenario) with well-calibrated model will be valuable but this study is far from it at this moment.

Consequently, I have to evaluate this manuscript as rejection. I am happy if the following comments would be useful.

Specific comments and Technical corrections

L85-87: Please explain how you dealt with "BIO" pool of the original RothC, too, here. You mention other four pools but not BIO.

L96-97: The original RothC uses monthly precipitation and open pan evaporation to calculate soil moisture condition. Did you change this part by using AWC? If so, please explain.

L98: What kind of soil properties did you estimate by visible-near infrared spectra.

L101-105: You must explain more about land cover here including definitions of cropland, modified or native razing land and native environments (which appeared in Figure 2).

L127-132: I did not understand this part.

L136: How did you relate evapotranspiration value with pan evaporation value? Explanation needed.

L150: Please show specific value of shoot to root ratio and show reference.

L151-155: Many assumptions here. Where did you get the value 1.25, for example? Please show references for each assumed value.

L157: How did you calculate 0.049?

L167: What does "each site" here mean? "73 sites" in L161? Or 4431 sites in L177? Explanation is not easily understandable.

L169-171: I think 100 years are too short to reach equilibrium. How did you set 100 years here?

L171-172: "from their initial values by a fraction of 1/100" is not clear explanation. From which value (minimum) to which value (maximum) for example? Please explain more in detail.

L182: Monthly variation?

L183-184: Why 10 Mg C ha-1 to exclude?

L184-185: This sentence should be in "Results".

L188-19: 100-years of future prediction generally uses future climate change scenarios. Why the authors did not do so? Did you use just current meteorological condition for future 100 years?

L190-191: Is 6 times greater C inputs achievable? This is very large amount so you have to discuss if such amount of organic matter could be available in terms of resource availability.

L195: Why 11-year moving average? Explanation needed.

L198: 100 years is not enough to reach equilibrium in many cases. How did you judge if it reached equilibrium or not? Explanation needed.

L213-214: I could not read median value from this figure.

Figure 3: Some of characters of horizontal axis are overlapped and not visible.

Figure 4: Title of figure is not easily understandable.

L246-247: please show data to support this sentence.

Figure5: TOC in left panels should be ROC. TOC=POC+HOC+ROC. Is this correct? Definition of vulnerability should be explained in Figure caption, too, even it is in main text, so that figure can be self-understandable.

L257-258: Why changes in stock under grazing and cropping will be similar if climate and soil texture have a dominant effect? Not understandable. Explanation is not enough.

L259-260; 261-263: This should be due to the difference of DPM/RPM ratio. Please add discussion on this.

L270-271: This sentence is not needed. Should be deleted.

L286: I did not understand the relationship between this sentence and sentences before and after.

L297-298: This comparison does not make sense because the area of each land use is different.

L304-306: So why you did not use more complete dataset like Viscarra Rossel et al. (2014, 2019)?

L306-308: I could not understand why this concluding sentence appears here. It is disconnected from sentences in this paragraph.

L313: I do not think this is "plausible" as mentioned above.

L313-315: You must discuss the reason of these difference among land use.

L316-318: You must discuss or explain why soil C become more vulnerable when soil C increases. Sentence of L317-318 does not say anything.

L327-329: You must explain more why this C input level was plausible. Explanation is not enough.

L330: I could not imagine how to "manage it locally". Explanation needed.

---

## Referee Comment (RC2) · Anonymous Referee #1 · 7 Aug 2020

This manuscript presents a simulation work on soil C dynamics using the RothC model over Australian croplands and grasslands. This topic is within the scope of the journal.

The manuscript has a strong potential, as it uses a large and continental-scale set of plant and soil data for model parametrisation, simulation and prediction. However, the manuscript suffers important issues of orientation of study objective, modelling and redaction, rendering the nice dataset not well valorised.

In the manuscript, the proposed framework that allows bridging dataset and the model plays a central role in driving the study's storyline (see LN1-3 as the beginning of Abstract, LN54-64 as the key sentences for knowledge gap identification in Introduction

and a whole Section 4.2 related to the framework). Too much emphasizing the framework makes the manuscript very technical, rather than scientific. First, there are no alternative frameworks presented as a control for comparison, so the advantages and drawbacks of the framework cannot really be validated. Then, the novelty of such a framework is unconvincing. There are numerous studies on soil carbon modelling performed at regional, national or bigger levels in literature. In such a kind of study, gathering, synthesizing, processing and standardizing large-scale climate, plant and soil datasets from diverse sources are usually common and necessary steps in the modelling process. The flow chart in Figure 1, as well as the associated discussions does not seem to be particularly special or innovative to what has been routinely done in literature. Therefore, selling a framework makes the manuscript scientifically weak and structurally unbalanced.

In parallel, the nice dataset over the continental scale could have been used to address very appealing ecological/agricultural questions, such as impacts of land-use and grazing on the long-term soil C fates. The manuscript indeed presents some figures (Figures. 3-5) including these treatments, but there are no scientific questions driven behind and no associated knowledge gap could be found in Introduction. Nor were these effects fully discussed in the current version of Section 4.1 of Discussion, which is, again, fairly technical and, in most of time, centred to the model. As one of the most famous pool-based models, RothC itself and the associated modelling skills are well-documented. In my opinion, given the nice dataset, it would have been more original to focus on specific questions about land management than on the model or a "framework".

The model's initialization procedure (LN160-163 and LN169-171) needs to be clarified too. It is well-known that the settings of initial relative sizes of soil C pools have a huge impact on the final outcomes. If I understand, at time 0, the authors used site-dependent (presumably observed?) carbon quantity with relative pools sizes corresponding to those at their theoretical equilibrium condition provided by the model,

right? It is not very clearly said in the text. Have the simulated C dynamics or changes ever been compared with those (presumably?) measured at the 73 sites from 1991 to 2000 (LN161)? This may be a good manner to check and validate the "equilibrium condition" hypothesis, which has been considered strong and untrue by increasing studies.

It is a very good idea to carry out an uncertainty analysis to test the impact of biomass DPM/RPM ratios on model results (LN166). However, the choice of biomass DPM/RPM ratio (LN184) is disputable. Despite some plasticity, a species' DPM/RPM ratio shall be quite stable depending on its taxonomical and functional features. For example, legumes which are richer in N and lower in C:N ratio shall have generally higher DPM/RPM ratios than grasses (e.g., rice, wheat. . .). But the authors' manner of choosing DPM/RPM ratio (". . . selected the DPM/RPM ratio based on the minimum deviation of TOC"; see LN 184) may risk picking unrealistic values for species. This is because the model's fit/bias is not only dependent on biomass DPM/RPM ratio, but also on the settings of relative soil C pool sizes at time 0 (whose influence on model fit is even much more important). Therefore, it would be more reasonable to choose species' DPM/RPM ratios according to the literature data on plant decomposition traits (even though they were not published in Australian contexts). An additional uncertainty analysis would always be appreciated to test the amplitude of impact of chosen DPM/RPM ratios with variance for a given setting of relative soil C pool sizes at time 0.

The removal of the 388 sites may need some more justifications. Why did these sites (not the others) yield such unrealistic values? Why 10 Mg C/ha as the threshold?

LN 209: Which model type did the authors choose for the test of the environmental factors? How do these factors cross or nest among each other? And how were they treated in the model? Additional information about this may be represented in Materials and Methods and Figure 6. When looking at the Figure 6, it is not surprising to see that Clay, MAT, MAP and PET stand out, as they are all directly or indirectly involved in the model as key inputs/parameters. What would be much more meaningful is to do the same test over the residuals (i.e., measured C changes minus modelled C changes

for the 73 sites 1991 to 2000). This helps see which environmental factors should be further taken into consideration by the model.

Overall, the manuscript tackles a timely and important question in the current research context. However, due to the unconvincing scientific orientation, unbalanced structure and ambiguous modelling procedures, I don't think the manuscript is mature enough to reach the journal's standard. I sincerely suggest the authors make good use of such a nice dataset and rework on the question and modelling processes.

---

## Author Comment (AC1) · 25 Sep 2020

**Response to reviewer 2 (R2): 'Simulation of soil carbon dynamics in Australia under a framework that better connects spatially explicit data with ROTH C'**

**Authors:** We thank R2 for taking the time to review our manuscript. Below, we provide our responses in blue text.

**R2:** The authors developed soil C calculating system which connects spatial datasets on meteorology, soil, land use and land management with the RothC model. They calibrated the RothC and predicted changes in soil C for 100 years with different soil management scenarios. I think this work is within the scope of this journal and potentially many of audience of Journal would be interested in. This was my first impression after quick read of paper. But after careful reading, I found some severe problems. First, explanation of what the authors have done is not enough throughout the paper especially in "Materials and methods" section. Frequent disconnection in logic between sentences made me difficult to understand what the authors really have done. Development of calculation system is great achievement, but it was difficult to evaluate the validity of the many assumptions in developing the system and future simulation procedure. This might be partly because of English skill but I think not only due to that. Significant re-writing of manuscript with English check by native speaker will be needed for this manuscript.

**Authors:** We thank the reviewer for noting that our work is in the scope of the journal. We regret to hear that he/she could not fully understand what we did. It would have been useful if the reviewer pointed out exactly which part(s) of the manuscript he/she found to be unclear or illogical. We do not see where our writing is with 'frequent disconnection in logic between sentences'. After re-reading our manuscript, we did not find any of those 'disconnections'. Such general statements are not useful because we cannot check them. To better respond, we need specific page and line numbers.

It isn't entirely clear what the reviewer means by 'Development of calculation system is great achievement...' but it sounds positive, so thank you. The comments around 'validity of the many assumptions' are confusing and inaccurate: the primary assumption that we made is that of equilibrium conditions of the current soil organic C for the baseline simulations. We believe that our description of the simulations under a standardised framework, which is described and depicted in Figure 1, is clear. Still, of course, we would have been open to improving our explanations and writing if the reviewer had

been more specific. Similarly, the comment around 're-writing' of the manuscript isn't particularly helpful. The reviewer needed to point out precisely where our English lacks for us to be able to respond. The corresponding author is practically a 'native speaker', and other colleagues who are proficient in English read the manuscript before submission. Of course, we could 'fine-tune' and 'tighten' some of our writing, but, we argue that, for the most part, our paper is grammatically correct.

**R2:** Second, future projection of 100 years generally requires the use of climate change scenarios but there is no description on this. I understood that the future projection in this study was conducted by using current meteorological data. This is curious.

**Authors:** We do not know where the reviewer got the impression that we ran 'future projections'. Please note that we ran long-term simulations, not projections with or without climate change scenarios. Our intent here was to look at the potential for soil C capture, not at the effects of climate on soil C stocks.

**R2:** Third, the setting of the changing amount of C input in future projection is not realistic. I do not think six times higher organic matter input is realistic scenario. It is natural that increasing C input result in higher soil C qualitatively of course. Quantitative estimation by using realistic scenario (both future climate change and management scenario) with well-calibrated model will be valuable but this study is far from it at this moment. Consequently, I have to evaluate this manuscript as rejection. I am happy if the following comments would be useful.

**Authors:** Again, what we did was not a future projection. More importantly, it would have been more useful if the reviewer provided some evidence to support why he/she thinks the range of different C inputs that we considered is not realistic. For example, C inputs six times larger than the baseline are possible with manuring. We selected the rates to represent a wide range of possible C inputs.

It is disappointing to read the reviewer's recommendation to reject our manuscript. Based on the comments made, it appears that his/her advice is based mostly on misunderstandings and misinterpretations that seem to stem from preconceptions.

We thank the reviewer for the specific comments. We respond to those next.

**R2:** L85-87: Please explain how you dealt with "BIO" pool of the original RothC, too, here. You mention other four pools but not BIO.

**Authors:** We could add the sentence as follows: "The BIO pool was initially set to zero (Sparling, 1992)."

**R2:** L96-97: The original RothC uses monthly precipitation and open pan evaporation to calculate soil moisture condition. Did you change this part by using AWC? If so, please explain.

**Authors:** Thank you for the comment. We did not modified the original routine that calculates topsoil moisture deficit from rainfall and open-pan evaporation. We used AWC to derive evapotranspiration (ET) from pan evaporation when a plant is present and then to calculate a potential biomass production that is water-limited using the ET. To clarify, we could revise as follows "The available water capacity (AWC) of the soil to a depth of 1 m is needed to modify evapotranspiration from pan evaporation when a plant is present and used to run a crop model (see below)."

**R2:** L98: What kind of soil properties did you estimate by visible-near infrared spectra.

**Authors:** We can revise as follows to clarify "The soil properties ..." to "The C fractions, clay content, and AWC ...".

**R2:** L101-105: You must explain more about land cover here including definitions of cropland, modified or native razing land and native environments (which appeared in Figure 2).

**Authors:** Thank you for the suggestion, we agree and can improve the definition of the selected broad land uses in section 2.3.2 as follows: "We defined cropping as land under broadacre crops. Modified grazing was defined as land used for livestock grazing on improved pastures with exotic vegetation cover. Native grazing was defined as land used for grazing on native pastures. Natural environments include the areas for nature conservation, indigenous uses, and other minimal uses."

**R2:** L127-132: I did not understand this part.

**Authors:** Here, we described how the dominant crop/grass species were determined based on the activity data derived by Unkovich et al. (2017). We could rewrite: "For each of the periods 1990–1994, 1995–1999, 2000–2004, 2005–2009, 2010–2014, we calculated the cumulative frequency by regime. We used it to randomly select the crop or grass species (both annual and perennial) through time with a probability approach. The probability to have a certain crop was dependent on the cumulative frequency assigned to each crop type and regime." Would this clarify?

**R2:** L136: How did you relate evapotranspiration value with pan evaporation value? Explanation needed.

**Authors:** Thanks, yes we could clarify as follows: "Daily evapotranspiration was estimated by multiplying pan evaporation with a ratio of soil water content over plant AWC and the maximum evapotranspiration by crop or grass."

**R2:** L150: Please show specific value of shoot to root ratio and show reference.

**Authors:** Thank you. We used a generic root-to-shoot ratio of 0.3, not a crop-specific value due to a lack of data for Australian conditions. However, we have added Bolinder et al. (1997) to the references to show this chosen value within the typical ranges.

**R2:** L151-155: Many assumptions here. Where did you get the value 1.25, for example? Please show references for each assumed value.

**Authors:** These assumptions were based on typical cropping and modified pasture management practices in Australia and are the ones typically considered in the APSIM model. According to the Department of Primary Industries and Regional Development (DPIRD), grazing pastures should be based on the pasture growth of dominant pastures (e.g. clover). Their optimum growth occurs at about 1400 kg DM/ha, with the recommended minimum of 1000 kg DM/ha, to maximise pasture production. We can provide the references as suggested.

**R2:** L157: How did you calculate 0.049?

**Authors:** Simply, the number represents the small amount of C inputs from sparse vegetation in arid and semi-arid climates. The monthly C inputs amount to roughly 0.5 Mg C/ha/year. According to Wang and Barrett (2003), which is now cited, a typical C production at these areas is relatively low, ranging from 0.1 to around 1 Mg Mg C/ha/year.

**R2:** L167: What does "each site" here mean? "73 sites" in L161? Or 4431 sites in L177? Explanation is not easily understandable.

**Authors:** These sites are the 4,431 sites, which should be clear from the context in which the sentence is written. However, we could be more specific and write "4,431 sites"'.

**R2:** L169-171: I think 100 years are too short to reach equilibrium. How did you set 100 years here?

**Authors:** The reviewer misunderstood. We initialised the model with measured C fractions to represent the pool structure of Roth C. We did not use a typical spin-up simulation to establish the relative size of the conceptual pools. The 100 years is for the baseline simulations assuming no change in environmental conditions and land management that would affect decomposition. In fact, we set the simulations to ensure that both dynamic pools were at equilibrium over the 100 year period. The reason we choose 100 year was because we wanted to make predictions over the this period, which corresponds to the Australian Emission Reduction Fund permanence period for carbon farming projects.

**R2:** L171-172: "from their initial values by a fraction of 1/100" is not clear explanation. From which value (minimum) to which value (maximum) for example? Please explain more in detail.

**Authors:** Please note that the amount of C inputs initially derived by the model was different for the 4,431 sites. For example, if the default value is 1 Mg C/ha, the default value would change by 1/100 at the first iteration. And then at the second iteration (and still in transition phase), the modified value at the previous iteration would change again by 1/100 and so on. We note that inadvertently, we had written 'a fraction of 1/100...', where in fact we simply meant 1%. 'We could change the sentence to "The monthly input of plant residues and farmyard manure changed from their initial values by 1/100, and then this step was reiterated from the modified values until equilibrium was achieved."

**R2:** L182: Monthly variation?

**Authors:** Yes, we calculated the range of variations in the simulated TOC stocks on a

monthly basis.

**R2:** L183-184: Why 10 Mg C ha-1 to exclude?

**Authors:** Our selection of this threshold was based on the National Carbon Accounting System (NCAS) dataset, where we found the range of yearly changes to be up to 10 Mg C. Thus, we selected this as the threshold. We described the conditions that need to be satisfied for modelled soil organic C reaches equilibrium. For these 388 sites, one of the dynamic pools, POC or HOC, failed to be constant with time. We could clarify the sentence as follows: "We considered 10 Mg C ha$^{-1}$ as the threshold based on the range of measured annual changes in TOC." We did write that the 388 sites were characterised by large TOC stocks (median 75.04 Mg C/ha), but we do not know why these sites had such large changes in the dynamic pools. We do not yet know whether or not these are unrealistic. One possibility is that the pool composition of large organic C stocks is not fully constrained by the decay rates and environmental factors (see Figure 6).

**R2:** L184-185: This sentence should be in "Results".

**Authors:** Thank you for the suggestion, but no, we need it in the materials and methods because we excluded these sites from further simulations.

**R2:** L188-19: 100-years of future prediction generally uses future climate change scenarios. Why the authors did not do so? Did you use just current meteorological condition for future 100 years?

**Authors:** We did not use future climate change scenarios because that is not the purpose of this particular manuscript. Yes, that is correct, we calibrated the model and then ran simulations to look at the effects of changing C inputs.

**R2:** L190-191: Is 6 times greater C inputs achievable? This is very large amount so you have to discuss if such amount of organic matter could be available in terms of resource availability.

**Authors:** This comment is similar to one the reviewer made previously. We performed the simulations using a realistic range of C input changes that correspond to a wide range of activities. We agree that manure addition might be unrealistic for most systems, however, it provides a feasible upper limit. We thank the reviewer for his/her opinion on the need to think about resource availability, but we do not believe that such commentaries will strengthen our argument or manuscript.

**R2:** L195: Why 11-year moving average? Explanation needed.

**Authors:** Moving averages are generally computed for environmental data such as climate to smooth out the decadal trend in the data. We used 11 years because it is generally thought that it takes around 10 years to capture meaningful soil C changes due to management changes.

**R2:** L198: 100 years is not enough to reach equilibrium in many cases. How did you judge if it reached equilibrium or not? Explanation needed.

**Authors:** This comment is similar to one made previously and we responded. The reviewer needs to understand that the model was site-specifically initialised with measured C fractions—no spin up simulations needed.

**R2:** L213-214: I could not read median value from this figure.

**Authors:** We apologise for that. We could add the median values to the caption of the figure.

**R2:** Figure 3: Some of characters of horizontal axis are overlapped and not visible.

**Authors:** That is strange. We cannot see the issue in our figures. Perhaps the editor can help?

**R2:** Figure 4: Title of figure is not easily understandable.

**Authors:** If it helps, we could change the caption to: "Changes in total, particulate, and humus organic C ..."

**R2:** L246-247: please show data to support this sentence.

**Authors:** We do not think that showing data on this would clarify or strengthen our point, and would simply be redundant. If the editor thinks it would help, we could provide climate maps to cross check with Figure 4.

**R2:** Figure5: TOC in left panels should be ROC. TOC=POC+HOC+ROC. Is this correct? Definition of vulnerability should be explained in Figure caption, too, even it is in main text, so that figure can be self-understandable.

**Authors:** Here, TOC also included the DPM and BIO pools. Since we do not discuss ROC we would keep TOC on the plot. We could do as suggested and include the C vulnerability equation in the caption: "The C vulnerability is derived by POC/(HOC + ROC).".

**R2:** L257-258: Why changes in stock under grazing and cropping will be similar if climate and soil texture have a dominant effect? Not understandable. Explanation is not enough.

**Authors:** Good point. We could clarify it to "..., possibly because of a similar range of

climate and soil texture that have a dominant effect of on the C inputs in these areas."

**R2:** L259-260; 261-263: This should be due to the difference of DPM/RPM ratio. Please add discussion on this.

**Authors:** We do not understand this comment. We described that the changes in TOC and POC followed a similar pattern under both cropping and grazing. Please note that the DPM/RPM ratio was optimised during model calibration so here the main driver for the changes would be largely due to changing the amount of C inputs.

**R2:** L270-271: This sentence is not needed. Should be deleted.

**Authors:** It isn't clear why the reviewer think the sentence should be deleted. We prefer to keep it, thank you.

**R2:** L286: I did not understand the relationship between this sentence and sentences before and after.

**Authors:** We do not see anything unclear in the sentences. The first discusses the site-specific estimation of the model parameters and the second suggests that our approach optimised both the amount and the quality of C inputs to maintain the current baseline soil organic C stocks. No change needed here

**R2:** L297-298: This comparison does not make sense because the area of each land use is different.

**Authors:** The reviewer misunderstands and must realise that both studies were conducted in Australia using the same land use classification: cropping regions, areas of modified and native grazing and natural environments are the same!

**R2:** L304-306: So why you did not use more complete dataset like Viscarra Rossel et al. (2014, 2019)?

**Authors:** Thank you for the comment. Yes, we could clarify by adding in section 2.3.1: "We selected a total of 4,431 out of 5,721 sites across Australia (Viscarra Rossel et al., 2019) (Figure 2). The selected sites were under the dominant land use, namely cropping, grazing of modified pastures and native vegetation, and natural conservation and protected areas. Native forests and production forestry were excluded because of a lack of simulation capacity."

**R2:** L306-308: I could not understand why this concluding sentence appears here. It is disconnected from sentences in this paragraph.

**Authors:** We do not understand the comment. In terms of the sentences in question, there is no 'disconnection'. The first, suggests why our ROTH C baseline estimates of the C stocks and composition differ somewhat from those produced by Viscarra Rossel et al. The second sentence expands, suggesting that unlike those previous estimates, the ones we present here, with ROTH C, can explain the soil processes that are important for estimating the baseline stocks of soil organic C and its composition.

**R2:** L313: I do not think this is "plausible" as mentioned above.

**Authors:** We thank the reviewer for his/her opinion, but it would be more useful if he/she could provide evidence to support his/her comment. Increasing C inputs by up to 3.5–12 Mg C/ha is entirely possible. For example, via management changes, e.g. manure addition. As with our previous response, we agree that manure additions might not be practically or economically feasible everywhere, but it does provide our simulations with an upper range. We do not see a problem with this.

**R2:** L313-315: You must discuss the reason of these difference among land use.

**Authors:** We did discussed this point, however, we could improve this discussion.

**R2:** L316-318: You must discuss or explain why soil C become more vulnerable when soil C increases. Sentence of L317-318 does not say anything.

**Authors:** We have discussed this point. Please see the main text (line 340).

**R2:** L327-329: You must explain more why this C input level was plausible. Explanation is not enough.

**Authors:** We responded to this comment already.

**R2:** L330: I could not imagine how to "manage it locally". Explanation needed.

**Authors:** We think that this aspect of our discussion is clear. However, if not taken in context, it could be challenging to understand. Paraphrasing our argument around L330, our work has shown that the baseline rate of C inputs into the active POC and HOC pools is site-specific. Therefore, soil management (e.g. via farm management practices like increasing residue retention rates) needs also to be local (i.e. site-specific), else we risk mismanagement and soil C loss. Further, locally derived (i.e. site-specific) C inputs are needed to identify soils that could potentially sequester C.

Please also note the supplement to this comment:
https://bg.copernicus.org/preprints/bg-2020-150/bg-2020-150-AC1-supplement.pdf

---

## Author Comment (AC2) · 25 Sep 2020

**Response to reviewer 1 (R1): 'Simulation of soil carbon dynamics in Australia under a framework that better connects spatially explicit data with ROTH C'**

**Authors:** We thank R1 for taking the time to review our manuscript. Our responses are below, in blue text.

**R1:** This manuscript presents a simulation work on soil C dynamics using the RothC model over Australian croplands and grasslands. This topic is within the scope of the journal. The manuscript has a strong potential, as it uses a large and continental-scale set of plant and soil data for model parametrisation, simulation and prediction.

**Authors:** We thank the reviewer for the comment and for acknowledging that our manuscript is suitable for Biogeosciences.

**R1:** However, the manuscript suffers from important issues of orientation of study objective, modelling and redaction, rendering the nice dataset not well valorised.

**Authors:** The 'orientation...study objective and redaction' of our manuscript are to use the ROTH C model initialised with measured C fractions under a framework that explicitly connects disparate datasets with the model, to simulate soil C change under different land uses across Australia. There is no clear indication of how the reviewer thought we should better 'valorise' our dataset, so we cannot directly respond to that. Below, we summarise our intentions and hope that the reviewer might now understand the value and novelty of our research.

- We demonstrate the simulation soil organic C across Australia with the ROTH C initialised with measurements of the particulate, humus, and resistant organic C (POC, HOC and ROC, respectively), under a framework that enables the synthesis, processing and standardisation of measurements and data, and predictions.

- We initialised the model site-specifically (across Australia) with the measured C fractions and optimised the DPM/RPM ratio (also site-specifically) because we believe that these are essential to accurately represent the baseline soil organic C stocks and composition across different land uses. This is crucial for the model to be used with confidence and for predicting changes in C stocks and the potential of soils for C sequestration. We have not seen this approach reported in the

literature.

- We showed that our simulations, with the model initialised as above, and using a 'standardised' modelling framework, accurately predicted the baseline soil C stocks and composition in the 0–0.3 m layer cross cropping, modified grazing and native grazing sites across Australia. Our predictions across natural environments were less accurate, but as we say in the manuscript, that was to be expected.

- We then used the model to perform a 100-year simulation and showed that with an annual increase of 1 Mg C ha$^{-1}$, the potential to increase organic C stocks, as well as the potential vulnerability to C loss, in Australian soils is smallest in soils under natural environments, larger under cropping and modified grazing, and the greatest in the soils under native grazing.

- Finally, we identified the soil and environmental controls on **the predicted changes.**

If our title or aims didn't capture well our intent, we can improve them.

**R1:** In the manuscript, the proposed framework that allows bridging dataset and the model plays a central role in driving the study's storyline (see LN1-3 as the beginning of Abstract, LN54-64 as the key sentences for knowledge gap identification in Introduction and a whole Section 4.2 related to the framework). Too much emphasizing the framework makes the manuscript very technical, rather than scientific.

**Authors:** We disagree with this comment. From our perspective, the flow of our argument is clear: we researched C dynamics across Australia using the ROTH C model, described the framework under which the research was performed, and the experiments and simulations. Finally we describe and then discuss our findings around the potential C increases in Australian soils under the different land uses.

The suggestion that the manuscript is 'very technical, rather than scientific' is a rather delusive statement about scientific research. Technologies/methodologies and 'science questions' are intimately linked. Yes, we agree that the latter are important, but we argue that how we arrive at the new science is equally important. It is critically important to explicitly describe the 'technical' aspects if we are to improve the quality of the science. This might be specially so in our case because of the complexity and scale of our research. We ran a model and simulated C dynamics over all of Australia, with thousands of measurements, with many continental-scale and different datasets and testing different aspects of the simulations.

In the literature, we often see studies that quite generally describe the model used, the input data and the simulation. This might be fine if the modelling is being performed at a single, or few sites where the datasets used are relatively small and originate from one or few sources. However, for large-scale, complex simulations one needs to know exactly how the various data are connected with the model. Additionally, and in support of our argument, we note that the development of simulation frameworks for reliable C predictions are recognised as a current topic of research (for example, see Smith et al., 2020). Therefore, we believe that the description of the framework is necessary.

Nevertheless, if the general perception is that the technical–science balance is not quite right, we would be happy to improve the title and aspects of the manuscript to emphasise the 'science questions'.

**R1:** First, there are no alternative frameworks presented as a control for comparison, so the advantages and drawbacks of the framework cannot really be validated. Then, the novelty of such a framework is unconvincing. There are numerous studies on soil carbon modelling performed at regional, national or bigger levels in literature. In such a kind of study, gathering, synthesizing, processing and standardizing large-scale climate, plant and soil datasets from diverse sources are usually common and necessary steps in the modelling process. The flow chart in Figure 1, as well as the associated

discussions does not seem to be particularly special or innovative to what has been routinely done in literature. Therefore, selling a framework makes the manuscript scientifically weak and structurally unbalanced.

**Authors:** We agree that the framework that we used isn't particularly special. However, we argue that explicit description of complex and large-scale simulations need to be undertaken under one such framework, and that it is critical that it is explicitly described: the datasets, how they were prepared and used, the experiments, the simulations, etc. Hence, we do not see why we would need to compare our framework to other frameworks.

We do not know of any other literature where a C model was initialised with measured C fractions with a dataset like ours and over a large scale. If there is other similar published work, we regret to have overlooked them, and will appreciate it if the reviewer made us aware of those. We believe that the quality and validity of our science is enhanced by explicitly describing the framework (including Figure 1). Our work addresses important scientific questions pertinent to Australian soils: e.g. how to represent the current soil C stocks and composition in different environments under different land uses; what is the quality and amount of C inputs required to maintain the current C levels; what is the potential to change soil organic C stocks with different C inputs.

**R1:** In parallel, the nice dataset over the continental scale could have been used to address very appealing ecological/agricultural questions, such as impacts of land-use and grazing on the long-term soil C fates. The manuscript indeed presents some figures (Figures. 3-5) including these treatments, but there are no scientific questions driven behind and no associated knowledge gap could be found in Introduction. Nor were these effects fully discussed in the current version of Section 4.1 of Discussion, which is, again, fairly technical and, in most of time, centred to the model.

**Authors:** We thank the reviewer for the comment about our data. We found the comment a little unclear. It seems that the reviewer acknowledges that our work contains important scientific results, but suggests that we clarify our intent and better describe the science questions and knowledge gaps (?). If this is the case, as we stated in our previous responses, yes, we can do that and also improve our discussion in section 4.1. However, we believe that there is scientific value in what we did (both in terms of the framework and the understanding gained), and we regret that our intent was misinterpreted.

**R1:** As one of the most famous pool-based models, RothC itself and the associated modelling skills are well-documented. In my opinion, given the nice dataset, it would have been more original to focus on specific questions about land management than on the model or a "framework".

**Authors:** Yes, the ROTH C model is well-known. We acknowledged this in the introduction. We also acknowledged previous research on its use in Australia. Our work is original, novel and unique because of how we used the model with the measured C fractions, other large-scale environmental and management datasets, and under an explicit framework to answer important questions about the potential of soil C capture under the main land uses in Australia.

**R1:** The model's initialization procedure (LN160-163 and LN169-171) needs to be clarified too. It is well-known that the settings of initial relative sizes of soil C pools have a huge impact on the final outcomes. If I understand, at time 0, the authors used site-dependent (presumably observed?) carbon quantity with relative pools sizes corresponding to those at their theoretical equilibrium condition provided by the model, right? It is not very clearly said in the text. Have the simulated C dynamics or changes ever been compared with those (presumably?) measured at the 73 sites from 1991 to 2000 (LN161)? This may be a good manner to check and validate the "equilibrium condition" hypothesis, which has been considered strong and untrue by increasing

studies.

**Authors:** We thank the reviewer for the comment. We can now see that section 2.3.4 may be difficult to follow. We can work to improve this. At each of the 4431 sites, we initialised the model's C pools with the measured C fractions, POC and HOC. Because our data represent a single time period, we made the equilibrium condition assumption using an independently collected dataset from Astralia's National Carbon Accounting System (NCAS). This dataset comprises C measurements at 73 sites across Australia mostly made twice, at intervals ranging from 2–20 years. Based on this assumption, we ran the model to optimise the unknown C inputs over the 100-year period. We thought that this was clear, but we can certainly improve the description in a revision. As we wrote in the manuscript, this equilibrium assumption may be untrue at some sites, but we do not actually know—there is no other data to confirm this. From the data that we have, and our understanding of our study sites, we believe that our assumption isn't unreasonable for sites under continuous, long-term cropping and grazing and those under natural environments. We are not aware of any study indicating transient conditions of temporal changes in soil C stock over large scales. So, we assumed equilibrium condition based on the NCAS data and validated the baseline soil organic C changes simulated by the model.

**R1:** It is a very good idea to carry out an uncertainty analysis to test the impact of biomass DPM/RPM ratios on model results (LN166). However, the choice of biomass DPM/RPM ratio (LN184) is disputable. Despite some plasticity, a species' DPM/RPM ratio shall be quite stable depending on its taxonomical and functional features. For example, legumes which are richer in N and lower in C:N ratio shall have generally higher DPM/RPM ratios than grasses (e.g., rice, wheat. . .). But the authors' manner of choosing DPM/RPM ratio (". . . selected the DPM/RPM ratio based on the minimum deviation of TOC"; see LN 184) may risk picking unrealistic values for species. This is because the model's fit/bias is not only dependent on biomass DPM/RPM ratio, but

also on the settings of relative soil C pool sizes at time 0 (whose influence on model fit is even much more important). Therefore, it would be more reasonable to choose species' DPM/RPM ratios according to the literature data on plant decomposition traits (even though they were not published in Australian contexts)

**Authors:** We tested six different values of the DPM/RPM ratio (0.67, 0.96, 1.17, 1.44, 1.78 and 2.23) to assess the sensitivity of the simulated TOC, POC and HOC to this parameter (section 2.3.5). Those values are not unrealistic. They are within the range of values used by Janik et al. (2002) in a sensitivity analysis (of NCAS data) performed under Australian conditions. Yes, the DPM/RPM ratio represents the potential decomposability of the incoming plant material and may be a function of its taxonomical and functional features at a species level. However, this ratio may differ within a species, for instance when representing different crop cultivars. Unfortunately, there aren't many studies that report direct species specific measurements of this ratio or how we could use plant biochemical properties (N and lignin) to link them to this ratio. It is realistic to account for possible parameter values and to check model performance. The performance of the model depends on the quality and quantity of C inputs, as well as the current (initial) soil organic C composition.

**R1:** An additional uncertainty analysis would always be appreciated to test the amplitude of impact of chosen DPM/RPM ratios with variance for a given setting of relative soil C pool sizes at time 0.

**Authors:** In this case, we think that a sensitivity assessment, like the one we performed, is more suitable for this particular analysis. The reason is that we used this to optimise the model. See Table S1 in the supplement for the range of C inputs and soil organic C stocks by the chosen ratios.

**R1:** The removal of the 388 sites may need some more justifications. Why did these

sites (not the others) yield such unrealistic values? Why 10 Mg C/ha as the threshold?

**Authors:** Based on the NCAS data, the range of annual change was found to be up to 10 Mg C and we selected this as the threshold. We described the conditions that need to be satisfied for the modelled soil organic C to reach equilibrium. For these 388 sites, one of the dynamic pools, either POC or HOC, failed to be constant with time. To clarify, we could add the following sentence: "We considered 10 Mg C ha$^{-1}$ as the threshold based on the range of measured annual changes in TOC." As described in the manuscript, the 388 sites were characterised by large TOC stocks (median 75.04 Mg C/ha). However, we do not know why these sites had such large changes in the dynamic pools—we cannot say whether or not these are unrealistic. One possibility is that the pool composition of large organic C stocks is not fully constrained by the decay rates and environmental factors calibrated in the model (see Figure 6).

**R1:** LN 209: Which model type did the authors choose for the test of the environmental factors? How do these factors cross or nest among each other? And how did they treat in the model? Additional information about this may be represented in Materials and Methods and Figure 6. When looking at the Figure 6, it is not surprising to see that Clay, MAT, MAP and PET stand out, as they are all directly or indirectly involved in the model as key inputs/parameters. What would be much more meaningful is to do the same test over the residuals (i.e., measured C changes minus modelled C changes for the 73 sites 1991 to 2000). This helps see which environmental factors should be further taken into consideration by the model.

**Authors:** We used the regression tree model, Cubist, which we described in section 2.3.6, lines 200–210. In a revision, we could provide more details on Cubist, but we note that this algorithm is well described elsewhere and we provided citations for the interested reader. We used Cubist to determine which environmental factors affected the **changes** in C stock of each pool (induced by different C inputs). This should be

clear from the subheading and text in section 2.3.6 'Empirical assessment of controls on the simulated C change'. Here, we are not interested in looking at the factors that affect the residuals from ROTH C. Therefore, in our case, because we modelled the changes, the analysis suggested by the reviewer isn't relevant.

**R1:** Overall, the manuscript tackles a timely and important question in the current research context. However, due to the unconvincing scientific orientation, unbalanced structure and ambiguous modelling procedures, I don't think the manuscript is mature enough to reach a reviewable condition. I sincerely suggest the authors make good use of such a nice dataset and rework on the question and modelling processes.

**Authors:** We thank the reviewer for acknowledging that our research is timely, but we disagree that our work has an 'unconvincing scientific orientation'. We hope that our responses above, have helped the reviewer to better understand the intent and novelty of our research. We also disagree that our modelling and simulation is 'ambiguous'—on the contrary, the thorough and explicit description of the simulations under the framework that we described, reports our approach in a transparent manner. We also disagree that the manuscript is too technical—it is essential to explicitly (and technically) describe the modelling approach for the simulations of soil C dynamics at such large scale to have real meaning. We do agree, however, that we can clarify and improve aspects of our writing.

The reviewer comments on the 'maturity' of our manuscript but does not specifically suggest why he/she thinks it is lacking. We hope to have addressed his/her main concerns regarding the scientific value of our work. We also hope to have clarified the relatively minor comments around the simulation procedure. Certainly, we can improve clarity in a revision.

The reviewer also suggests that we 'make good use' of our dataset, but he/she does not say what a better use might be. Thus, the purpose of the comments isn't' clear. Our

manuscript describes, for the first time, simulations of soil organic C across Australia with the ROTH C model site-specifically calibrated with measured organic C fractions and sensible DPM/RPM values across different land uses. Using the well calibrated model, we demonstrated the potential effects of changing C inputs on the changes in soil organic C stocks and its pools. We found that with an annual increase of 1 Mg C ha$^{-1}$, the potential for C sequestration, as well as the potential vulnerability to C loss, in Australian soils is smallest in soils under natural environments, larger under cropping and modified grazing, and the greatest in the soils under native grazing. The simulations across Australia were performed under a framework that establishes a much-needed connection between measurements, datasets and the model. It enabled consistent processing of measurements and datasets from different sources, and standardisation and configuration of the model for calibration, verification, and prediction.

Finally, although we disagree with many of the reviewer's comments, we see that we can improve our manuscript by clarifying our intent and better emphasising the significance of our findings. For this, we are grateful.

Please also note the supplement to this comment:
https://bg.copernicus.org/preprints/bg-2020-150/bg-2020-150-AC2-supplement.pdf

---

## Author Response (AR1)

**Major revision: 'Simulation of soil carbon dynamics in Australia under a framework that better connects spatially explicit data with** Roth C'

Dear authors

The handling editor decided to reject your paper following the recommendations of two referees. This decision was overturned by the Chief Editors following your appeal to the editorial board. I was asked by the editorial board to handle your manuscript.

I am happy to invite you to submit a suitably revised version of your manuscript, considering the comments by the two referees and my comments. Your revised manuscript will be sent out again to review.

**Authors:** We thank the Editor, Dr Joos, for taking the time to review our manuscript, the two reviews and our responses to the reviewers. Below, we describe the major revision of our manuscript, which are as the editor suggests and considering the the two previous referee's comments and our responses.

Please carefully consider the comments made by the two reviewers and change your manuscript to address these points where appropriate.

**Authors:** Thank you, we have done so.

I suggest that you add a few lines early on in the manuscript to clarify the setup of your sensitivity simulations with altered carbon input to soils under constant climate and provide an explanation/justification on the range of input fluxes selected to address the second and third main point of referee #2. In this context, please consider replacing the word "prediction/predict" with "simulations/simulate" or similar to avoid the impression that results represent a prediction of the future evolution of soil C in Australia.

**Authors:** Thank you for the suggestions. We addressed these points by: (i) reformulated our aims to emphasise the site-specific initialisation and optimisation performed; (ii) clarifying that the simulations to assess potential increases in C stock were made using constant climate (we made this clear throughout the manuscript), and a plausible range of C inputs; and (iii) emphasising that we wanted to test a wide and representative range of C inputs. Regarding

this last point, in the revision we added: "These rates were selected to represent a wide range of C input levels that would be either physically achievable or manageable (e.g. manure addition) (Maillard and Angers, 2014)." Please note that animal manure was used as fertilisers by 11% of agricultural businesses worldwide. About 2.1 million Mg of animal manure were applied in 2011-12. Also, we include a reference to Maillard and Angers (2014) who, in their global meta-analysis, report manure addition ranging from 0 to around 400 Mg in crop management systems.

Reviewer #1 perceived your work and presentation as very technical. You kindly offered to further emphasize the science questions in your reply to reviewer #1. It would, in my opinion, indeed be useful to provide a few additional lines further describing the context and motivation for your model framework and your sensitivity simulations in the introduction, thereby complementing the context and motivation already provided. I may be wrong, but I got the impression that one motivation of this study is to prepare the ground for further application of your model (i) in assessments of Australian National Greenhouse Gas Inventory (line 50) used for the reporting under the UNFCCC process (?) and (ii) to pay farmers for efforts to stimulate extra carbon transfer to soils (line 51). I also interpret your text on line 51, that your model may be used to estimate the change in soil carbon stock over a 100-yr period under constant climate in response to measures by farmers, and in turn, this estimate is used to pay farmers. In any case, I would welcome further motivation in the manuscript why you are performing this model exercise, e.g., also building on your answer to reviewer #1 on page C5 in the reply ("Our work addresses important scientific questions pertinent to Australian soils: e.g. how...").

**Authors:** We appreciate the suggestions. To address these points, we (i) improved out title to 'Simulation of soil carbon dynamics in Australia with ROTH C' — thus removing the said misleading focus on the 'framework'; (ii) re-formulated our aims to better describe the significance and innovation of our work—as per our responses to the referees, and (iii) clarified the motivation of our work, according to our previous responses to reviewer #1. Thus, in the revision, we write: "Here, we report on simulations of the organic C stocks in Australian soils

with ROTH C using a standardised approach that synthesises and processes measurements and data for prediction at a correct scale. Our motivation for developing this research is to help answer questions around soil C that are pertinent to Australian soils and ecosystems under different land uses and management. Our aims are to: (i) derive baseline estimates of soil organic C stock and composition by site-specifically initialising the model with measurements of POC, HOC and ROC and an optimised ratio of decomposable plant material (DPM) to resistant plant material (RPM), which represents the decomposability of incoming biomass, (ii) simulate over a 100-year period, with constant climate and a plausible range of C inputs, the potential to increase organic C stocks as well as the potential vulnerability to C loss across Australia, and (iii) to identify the soil and environmental controls of the change in soil C stocks.".

Reviewer #1 noted that section 2.3.4 is unclear. The description of your iterative procedure to estimate soil C input in section 2.3.4 should indeed be improved. What do you mean exactly by "We tested six different values of DPM/RMP ratio ... We then perform the simulation iteratively ..." Did you run for each value of DPM/RPM an iteration to estimate C input for a given DPM/RMP value? The sentence on l. 169/170 is unclear: "We ran the model for 100 years [ ] until equilibrium conditions occurred." Maybe you want to say that you run the model several times for 100 yr with slightly different C input until equilibrium is achieved? Does this also imply that the TOC inventory at the end of the iterative procedure is the same as prescribed at the beginning of the iteration (apart from the bacterial pool? Where the soil pools initialized again with the measurements after each simulation in the iteration? Please describe your simulation protocol more precisely.

**Authors:** We agree that our description was unclear. We have improved the description of these methods and are confident that in the revision, our description of these methods is clear. We now write: "We tested six different DPM/RPM ratios (0.67, 0.96, 1.17, 1.44, 1.78 and 2.23) to estimate baseline C inputs and to assess the sensitivity of the simulated TOC, POC and HOC to this parameter... For each DPM/RPM ratio, we run the simulations at each of 4,431 sites for 100 years. Specifically, for each ratio at each location, we performed the

simulations iteratively by re-initialising the POC and HOC pools with the measured C fractions and with a change in the monthly input of plant residues and farmyard manure equivalent to 1/100 of their initial values. This was repeated 1000 times or until equilibrium was achieved. We considered only monthly C inputs in the simulations. The weather data used in the simulations represents the conditions of the baseline period between 1991–2010, which were repeated over the 100-year simulation...".

Regarding the equilibrium assumption, it is not clear to me why you need to estimate soil input iteratively. Could this not be done by setting C input equal to all C loss fluxes?

**Authors:** This is a misunderstanding. The wording was unclear, we agree. We improved the description of these methods and hope to have clarified the procedure (see above). The simulations were performed for each DPM/RPM ratio at each location for 100 years. For each ratio at each site, the model was run iteratively by re-initialising the C fractions with the measurements and by changing the C inputs slightly, by 1/100 from their original values. This run up to 1000 times or until the model reached equilibrium. In ROTH C, when the ratio of DPM/RPM (i.e. the quality of the organic matter) is fixed, iterative adjustments to the amount of C inputs is needed because we do not know all C losses that will occur and we don't know the C inputs needed to compensate the losses.

In general, I found your proposed text modifications appropriate in response to the comments of referee #2. Please implement these changes when revising your manuscript for re-submission. Please also consider whether additional clarifications may help to avoid misunderstanding and address comments where you did not indicate any action in your response

**Authors:** Thank you. We have implemented the changes in the text as proposed in our previous responses and have made the additional clarifications to prevent misunderstanding.

I am looking forward to receiving your revised manuscript. Thank you for submitting your work to Biogeosciences.

Yours sincerely,

Fortunat Joos

Further minor comments:

L80: Please provide the reference state (Temp, soil water veg. cover, ..) for the specified values of the decomposition rate coefficients.

**Authors:** The reference state is the average conditions at Rothamsted, which was reported by Jenkinson and Rayner (1977). We have added the following sentence and listed this reference: "Its reference state for the decomposition rate constants was reported by Jenkinson and Rayner (1977)".

261/262: please clarify this sentence. It is unclear why an increase of 39% is larger than an increase of 59%?

**Authors:** Apologies for the confusion. We have revised the sentence to "The soil under native grazing was the most vulnerable with the increase in POC (35%) and HOC (59%), showing that the labile POC increased more proportionally than that of the other land uses.".

Figure 3: I had the same issue with overlapping labels as referee #2. Please check your figure and, perhaps, select slightly smaller fonts.

**Authors:** We have corrected the figure.

References

Maillard, É. and Angers, D.A. (2014) Animal manure application and soil organic carbon stocks: a meta-analysis. Glob Change Biol, 20: 666-679. https://doi.org/10.1111/gcb.12438

MLA (2002) Safe Use of Manure and Effluent - A Technical Users Manual, Published by Meat & Livestock Australia (MLA) Limited. ISBN 1 74036 362 0.

---

## Referee Report (RR1)

**Review of MS: "Simulation of soil carbon dynamics in Australia under a framework that better connects spatially explicit data with ROTH C " by J. Lee *et al.***

The authors have identified a genuine (and insufficiently noted) problem, i.e. that there is a general disconnect between models and observations of soil carbon dynamics. Their approach to resolve this is pragmatic, making use of a long-standing model (RothC) whose design, and (relatively low) level of complexity, are appropriate to the task. After site-specific initialization and some calibration, they were able to show very good agreement between observations and simulations. They then used the model to answer an important practical question regarding the potential for changes in land management practice to deliver carbon sequestration benefits.

The manuscript is generally well written and clear in its statements of objectives, assumptions and methods. There are many open issues about how best to model soil carbon dynamics, given that we now know that the conceptual categories used by this generation of models do not, in fact, correspond to chemically distinct classes of compounds, but rather to different degrees of physical protection from microbial attack. However, like the authors, I am not convinced that any of the recently published alternative formulations provides a useful way forward for applications of this kind. Meanwhile, work like this needs to be done, while in practice very little of it is being done anywhere. The originality of this research thus does not lie in any particular advance in modelling or theoretical understanding, but rather in the way it uses an established modelling framework to answer pressing real-world questions underpinned by a sound observational basis.

I would like to raise just one issue about the availability of data and codes. Today, in my view, it is no longer acceptable to make the underpinnings of a scientific paper available only "on reasonable request" – which leaves it open to the authors to deny access. This information should instead be made available via a public repository, thus greatly increasing the potential utility of the research as well as making the results open to alternative analytical approaches.

Colin Prentice

---

## Author Response (AR2)

**Response to Associate Editor**

**AEditor:** Your manuscript was meanwhile assessed by three new reviewers. All three reviewers note that your work is of high technical quality and of importance but request improvements before a potential publication. The three reviewers offer points of criticisms and very valuable advice on how to further improve the manuscript. I ask you to carefully consider the issues raised by the reviewers and to prepare a revised manuscript and a point-by-point reply to the review comments. The manuscript will then be sent to the two reviewers that asked for major revisions.

**Authors:** Thank you for recognising the technical importance of our work. We have revised the manuscript and provide point-by-point responses to all of the comments made. Please note that we also improved the title of the paper to better represent what we did. The new title is 'Assessing the response of soil carbon in Australia to changing inputs and future climate change using a consistent modelling framework' . We note that we have a new co-author in the manuscript. The the first author of the manuscript, Dr Lee, left our group and so we had reduced capacity to complete the revision. Dr Mingxi Zhang helped to perform the new simulations that account for climate change.

**AEditor:** Reviewer #3 asks that the underpinning data and codes are made publicly available. It is becoming more and more praxis and also requested by Biogeosciences to publish input data sets, results, and codes on a public server, preferentially with a doi attached. I ask you to follow this practice to the extent this is reasonable.

**Authors:** The research reported in this paper is the first of three planned publications from an on-going project that started three years ago. There's been significant investment in the collection and collation of data, the laboratory and sensor measurements, the development of the various code to implement the framework and the simulations, etc. Until we have completed the project and reported our planned outputs, it isn't sensible for us to release the data or code. Of course, we are always receptive to collaborations and would support innovative uses of the data and code, under specific agreement, depending on the request and

situation. Therefore, at this stage, we'd prefer to keep the statement as 'on reasonable request to the corresponding author'. We provide a more complete response to referee #3, below.

**AEditor:** Reviewer #4 raises three important scientific issues. (i) Estimating the effect of changing inputs on SOC stocks over a 100-year time horizon without consideration of climate change appears not meaningful. This is certainly a valid criticism as global warming is continuing. A relatively small effort, with results for a small number of climate scenarios and a 'mean' input, might inform the reader about the sensitivity of your results to different climate conditions. These additional results may be presented in an appendix and briefly discussed in the main text or presented in the main text. (ii) The reviewer also recommends to discuss shoot-root ratios under altered input and to compare these ratios to literature values as an overall quality control. (iii) Finally, reviewer #4 notes that some of the input scenarios are partially unrealistic. This point should at least be reflected in the discussion.

**Authors:** We have revised the manuscript as suggested. (i) We run additional simulations that account for climate change. These are described in a new section '2.3.6 Simulation: the potential for C sequestration under a changing climate' and the results are reported in the new section '3.3 Effect of changing climate on soil organic C' and new Figure 6. Of course, the new results are also discussed in the Discussion section. (ii) We provide a response and clarification to the comments around the shoot-root ratios, please see below. Please note, in our framework, optimising C inputs for the baseline would be less sensitive to this parameter as we have tuned the amount and quality of plant-derived C inputs. (iii) We have revised the C input scenarios by limiting these to $2 \times$ the baseline C inputs, and have revised the relevant sections of the manuscript as well as the figures.

**AEditor:** Reviewer #5 appreciates the technical aspects of your work but calls for an improved presentation. The reviewer critiques vague and subjective statements, and the lack of a clear definition of the problem, the outcomes, and the novelty. Please consider these comments very carefully and revise your manuscript accordingly.

**Authors:** We could understand most of the points made by the referee and we have

responded and revised the manuscript accordingly. As suggested, we have removed subjective expressions from the script.

**AEditor:** Please follow the guidelines of the journal when preparing your response: The author's response in case of "minor" or "major" revisions must be submitted as one separate *.pdf file (indicating page and line numbers), structured in a clear and easy-to-follow sequence: (1) comments from referees/public, (2) author's response, and (3) author's changes in manuscript. Regarding author's changes, a marked-up manuscript version (track changes in Word, latexdiff in LaTeX) converted into *.pdf and combined with the author's response should be provided.

**Authors:** We will do so. Thank you for the information.

**Authors:** We thank the three referees for taking the time to review our manuscript. Below are our responses that also indicate the specific revisions made.

**Response to reviewer 5**

**Referee:** Abstract: The opening of the abstract is hard to follow, I can imagine any number of "disconnects" not disconnections, but the abstract doesn't state the problem being addressed. I also think this is a technical issue and put a modeling issue before the science question. The abstract should begin with a clear statement about what the authors are actually studying. Something like "Soil organic matter is a key reservoir for carbon, and its vulnerability to global change depends also om many anthropogenic management activities"..

**Authors:** We have revised the opening of the abstract to "Land use and management practices affect the the variability of soil organic carbon (C) to global change through associations among climate, vegetation and soil. Soil process models are useful tools to simulate C dynamics, but it is important to bridge any disconnect that exists between the data used to inform them and the processes that they depict."

**Referee:** This is pretty generic: "However, a better understanding of soil organic C dynamics is needed to determine the size of the soil C pool accurately and to assess the potential for

those opportunities". Can you be more specific about why better understanding is needed? What are the big uncertainties? This statement can be said about anything, but without specifics, why bother saying this? This is an opportunity to be specific, not generic.

**Authors:** We revised the sentence as follows: "However, these opportunities depend on regional interactions between soil, climate, land use and management ... A better understanding of the effect of these interactions on soil C is needed to assess the potential for those opportunities."

**Referee:** Is science a popularity contest? Is this really what you mean to say? Other reasons for their continued popularity might be that there is ample documentation on them; they are relatively simple and general and are therefore also well understood. I would think, and it would certainly strengthen this paper if Roth-C and Century were not popular because they are simple and easy to use but because they are well-tested, accurate and robust (which I believe they are), and I believe they're little evidence that more complex models are more accurate for this type of study.

**Authors:** Of course, science is not a popularity contest. We meant that these models are the most commonly used for the reasons that the reviewer mentioned, which is the reason we used this particular model in our study. To reflect this, we have removed "for their continued popularity" and revised the sentence to "Other reasons for continued application of these models might be that there is ample documentation on them; they are relatively robust and well tested and are therefore also well understood."

**Referee:** A key part of this paper is the integration of the ROTH-C model and data, yet the introduction is strangely vague on this subject. What deficiencies arise from poor data synergy, and what approach is proposed? The paragraph around Line 60 is very superficial.

**Authors:** We disagree that our introduction is vague. This section states the challenges with soil C simulation modelling and cites relevant literature. The reviewer should note that the issues are not restricted only to the ROTH C model. In the introduction, we have a paragraph, starting with "Simulation of soil C dynamics with biogeochemical models can be challenging."

Then, we indicate a direction that we intended to follow, and then the last sentences in the same paragraph we write (about frameworks for simulation modelling): "Their development should also allow for their efficient updating, with new measurements, data and models, as they become available ..., and enable a more systematic approach for calibration and validation, making simulations more reliable and reproducible." Note also that the next paragraph describes our motivation and aims, which clearly state our proposed approach.

**Referee:** Figure 1: How is this different? The input data sets look very standard and can the figure show what's novel in this approach? Eg: remote sensing, spectroscopy, etc

**Authors:** How is this different to what specifically? We believe that the figure is important because it provides the reader with an easy-to-follow diagram that shows the steps that are included in our framework. The novelty of our approach is the integration of different data (from remote, proximal sensing, new data products, etc), which represent different scales and resolutions, and how they were processed, simulated, aggregated, etc.

**Referee:** I always discourage statements like the following: This gave us confidence in the performance of the model in Australia. What does "confidence" mean? Good enough for a paper, good enough to make management recommendations with financial conbsequences? I much preferred statements about the likely uncertainty, and openness about where and when the model could fail! Confidence is like belief, it has very limited place in a scientific paper!

**Authors:** We revised or deleted the word "confidence" in the ms, except when referring to confidence intervals. Specifically, in the abstract, we've revised "improve confidence in the model's estimates" to "improve the model's estimates". In the section 3.1, we've deleted the last sentence, which is "This gave us confidence in the performance of the model in Australia.". And in the section 4.1, we changed the relevant sentece to "... we empirically assessed how well the baseline simulations matched the model's corresponding dynamic pools, which suggest that the model is able to represent various Australian soils. "

**Referee:** Figure 3: these model fits are SO good they raise concern, when simulating so notoriously variable a quantity as SOC, some discussion of why and whether that would be

expected away from the calibration sites! In the caption, it states after optimization but it would be very easy for a casual reader to assume this is the likely model accuracy, especially after the "confidence statement".

**Authors:** Our manuscript describes the methods and steps that we took for the model optimisation and we reported the sites where the optimisation failed. We also hope now there is no confusion arising as we addressed the concerns over the word, "confidence".

**Referee:** Figure 4 and 5 are very nice. The estimate of importance in Figure 6 is quite unclear, and could be expanded or even defined in the caption.

**Authors:** We have added "The importance of each soil variable was assessed based on the usage of each individual variable in the rule conditions and the model for Cubist." to the caption, as suggested.

**Referee:** Around Line 305: I suggest avoiding "performed well" "additional confidence" and replace all of these statements that are subjective with explicit statements of model skill. Statements about "performing well" or "confidence" should be reserved for the very end of the discussion where the use of the model and output for managers is mentioned. In the results and discussion, focus on skill metrics, not subjective value statement.

**Authors:** We revised the use of subjective expressions throughout the manuscript. We've changed the sentence on line 310 to "The model explained 73–98% of the variation in the size of the C pools in soils that are under cropping and 86–98% of that under grazing, while the simulation under natural environments ..."

**Referee:** This statement remains difficult for me. What is the disconnect? By this point statements like "Soil measurements are very local and sparse, yet soil information is required at farm to national levels". Not "disconnect". There are specific challenges, not well-sumarized by "disconnect". This whole paragraph starting section 4.2 is redundant with earlier statements and adds no detail. What do these data sets add? I can see the authors grasping for a point here, but this entire section could be summarized by "Incorporating both local and mapped information enables the model to be run for large areas, or with more specificity for

particular sites". This has been done many times with ROTH-C (and Century)—these authors have developed a particular coding and data management approach, but it is very similar to gridded Century with its schedule files.

**Authors:** In this study, among many challenges, we are focusing on two challenges: 1) a lack of details in the previous approaches and 2) a need for a framework that better incorporate new measurements to simulate soil C dynamics depending on the specific choice of scales. We understand the comment.

**Referee:** Future needs is so generic as to be useless

**Authors:** We disagree with this comment. The points that we made are specific to the the framework built with the ROTH C model by listing what's still missing that requires in the workflow of the framework, as well as the direction that we think should take place for further model improvements in the near future.

**Referee:** The conclusions are repetitious with early text and could be strengthened significantly. There is a functional disconnection between measurements, data and biogeochemical models (Blankinship et al., 2018), but by simulating under a framework, like we did here, we can bridge that disconnect.

**Authors:** Thank you for the comment. We have done as suggested.

**Response to reviewer 4**

**Referee:** I reviewed the revised version of the MS, also taking note of the previous reviews and changes that were made to the new version.

This is a very interesting, technically demanding study where the controlling parameters input rate and input quality for soil carbon stocks in Australian ecosystems were evaluated by using RothC and an elaborated set of external data. The overall structure of the work, its general quality, and the fact that it seamlessly follows a number of previous Australian studies on the same topic, where different data sets and methods have been developed that were also used here, is appreciated.

**Authors:** Thank you.

**Referee:** In line with a previous comment, I see a major drawback in that the authors did not use climate change (CC) scenarios for calculating the effect of increasing C inputs to SOC stocks. Considering the well documented effect that CC will have on future SOC in previous modeling approaches from other regions, it seems not plausible to rely on a repetition of time windows of past climate to simulate the response of SOC to changing conditions which, by nature, can only take place in the future. I therefore suggest that the authors use CC scenarios for evaluating the effect of higher C input on SOC storage. In this context, also the discussion starting in line 362 could be extended towards climate as an important determinant of long-term SOC changes.

**Authors:** We have now run the simulations considering projected future changes in climate (1.5, 2 and 5.0°C) over a 100-years. The relevant sections of the paper were revised accordingly: New methods section '2.3.6 Simulation: the potential for C sequestration under a changing climate'; new results section '3.3 Effect of changing climate on soil organic C'; new Figure 6 and also revised the abstract, discussion and conclusions.

**Referee:** Estimating or simulating belowground input is crucial for every SOC modeling study. On pages 6-7, the approach used for estimating these inputs is described. A fixed shoot:root ratio is considered to get the belowground part from the crop model (whether or not shoot:root ratios vary for a specific crop depending on external conditions such as management is a matter of debate in the literature [e.g. Hirte et al. 2018], but such a simplified approach seems justified given the poor availability of belowground input data from experiments). The authors then change monthly inputs (page 8) to better understand the response of the SOC pool to it. In consequence, shoot:root ratios will change as well. I suggest to display and discuss the resulting shoot:root ratios, as they provide a means of quality control to the overall approach. Reviews such as the one from Bolinder et al. (2007) might be helpful to put the derived ratios into context.

**Authors:** We appreciate this comment. The shoot:root ratios are fixed as a constant over

time for both annuals and perennials, although we fully acknowledge the ratios tend to vary by species, management practices, location, time, etc. This is because, in this framework, we use a relatively simple plant model to set the starting amount of C inputs from plant residues and manures as a function of temperature and soil-water conditions so that we could consider different monthly growth rates of the plant. As this model is not directly linked to the soil organic matter routine of the ROTH C model, the routine is not dynamic and does not allow back-calculation of the resulting shoot:root ratios. In addition, their short-term variation would have a little impact on the longer-term soil C baseline when the model is optimised. Nevertheless, we point out the need of more plant properties data because of the need to verify the baseline C inputs by land use. Such data might be of use to improve the overall optimisation process within the framework.

**Referee:** The selected range of inputs of between 0.25 – 6 times the equilibrium input seems wide, and Fig. 4 indicates inputs of up to 10 t C ha-1 per year. Even though the authors acknowledge that allocating inputs to sites/soils where higher additional storage might be achieved (line 360) is aimed for, an increase by more than 2-fold seems highly unrealistic given issues of transport or nutrient input. For comparison, a recent study from the temperate zone came up with an estimate of c. 3.7 t C ha-1 for croplands and grasslands (Jacobs et al. 2020). Therefore, I consider these input scenarios as partially unrealistic and suggest to downscale them a bit.

**Authors:** We now report the simulations with a range of inputs of up to $2 \times$ baseline C inputs, and revised the relevant sections of the manuscript and figures.

**Response to reviewer 3**

**Referee:** The authors have identified a genuine (and insufficiently noted) problem, i.e. that there is a general disconnect between models and observations of soil carbon dynamics. Their approach to resolve this is pragmatic, making use of a long-standing model (RothC) whose design, and (relatively low) level of complexity, are appropriate to the task. After site-specific initialization and some calibration, they were able to show very good agreement between

observations and simulations. They then used the model to answer an important practical question regarding the potential for changes in land management practice to deliver carbon sequestration benefits.

**Authors:** We thank the reviewer for his/her understanding of the motivation of our research and our approach.

**Referee:** The manuscript is generally well written and clear in its statements of objectives, assumptions and methods. There are many open issues about how best to model soil carbon dynamics, given that we now know that the conceptual categories used by this generation of models do not, in fact, correspond to chemically distinct classes of compounds, but rather to different degrees of physical protection from microbial attack. However, like the authors, I am not convinced that any of the recently published alternative formulations provides a useful way forward for applications of this kind. Meanwhile, work like this needs to be done, while in practice very little of it is being done anywhere. The originality of this research thus does not lie in any particular advance in modelling or theoretical understanding, but rather in the way it uses an established modelling framework to answer pressing real-world questions underpinned by a sound observational basis.

**Authors:** Thank you.

**Referee:** I would like to raise just one issue about the availability of data and codes. Today, in my view, it is no longer acceptable to make the underpinnings of a scientific paper available only "on reasonable request" – which leaves it open to the authors to deny access. This information should instead be made available via a public repository, thus greatly increasing the potential utility of the research as well as making the results open to alternative analytical approaches.

**Authors:** We appreciate the reviewers comment, and generally, for single-output projects, we would agree as we also hold a similar view. However, in this case, currently, full release of all of the data and code isn't 'straight-forward'. The research reported in this paper is part of a project started three years ago and the project is on-going. There's been significant investment

in the collection and collation of data, the laboratory and sensor measurements, the development of the various code to implement the framework and the simulations, etc. This paper represents the first of three publications from this project. Until we have completed the project and reported our planned outputs, it isn't sensible for us to release the data or code. Of course, we are always receptive to collaborations and would support innovative uses of the data and code, under specific a agreement that will not hamper our research, which will depend on the request and situation. Therefore, at this stage, we'd prefer to keep the statement as 'on reasonable request to the corresponding author'.